# Functional synaptic connectivity shapes spine stability in the hippocampus

Cynthia Rais [1] & J. Simon Wiegert [1,2] ✉

Synaptic connections between neurons determine the flow of information in the brain. Changes in synaptic weight, along with synapse formation and pruning, reshape the functional connectivity of neural circuits—key mechanisms underlying learning and memory. However, the relationship between functional strength and the structural dynamics of individual glutamatergic synapses in the living mammalian brain remains poorly understood. Specifically, how spine morphology and stability relate to functional adaptations is unclear. Here, we repeatedly recorded excitatory postsynaptic calcium transients in single postsynaptic spines of CA1 neurons in response to optogenetic stimulation of presynaptic CA3 cells in awake, head-fixed mice for over 2 weeks. We found that functional connectivity predicted both the structural stability and spatial proximity of synaptic inputs. Spines with large responses exhibited larger volume and higher stability compared to unresponsive spines. Over time, responses were highly variable at individual synapses, but stable at the dendritic level, suggesting that dendritic branches receive stable input despite large fluctuations at individual synapses.

During learning, experience-dependent activity induces synaptic plasticity, modifying synaptic strength[1-3]. Most excitatory inputs in the brain form synapses on dendritic spines[4], where changes in synaptic strength are often accompanied by structural modifications[5-8]. This has led to the hypothesis that memory traces are stored at synapses[9,10] and, by extension, in dendritic spines. Among these, the Schaffer collateral/commissural synapses connecting the CA3 and CA1 regions of the hippocampus are considered as prototypical small glutamatergic synapses in the central nervous system. These synapses are crucial for long-term synaptic plasticity, which underlies the formation and consolidation of hippocampus-dependent declarative and episodic-like memories. The dynamic nature of memory formation and updating, particularly in response to a constantly changing environment[11] necessitates continual restructuring of neural networks[6,12]. This need for plasticity contrasts with the need for memory stability, raising a fundamental question: how does the brain maintain stable memory storage despite the inherent instability of synaptic connections? Several candidate mechanisms have been proposed[13], but the factors determining the longevity of dendritic spines remain only partially understood.

In the hippocampus, particularly in area CA1, the lifetime of dendritic spines has been tracked over the course of several days in order to assess the stability of synapses and their role in storing memory traces. However, findings remain inconsistent[14-17] with reported spine survival rates ranging from high stability to nearly complete turnover within a few weeks[17]. Many of these studies relied on repeated anesthesia, which is known to affect spine stability[15].

A further limitation of previous studies is that they focused on spine morphology without directly linking structural changes to synaptic function in vivo. Anatomical studies alone cannot reveal presynaptic input properties or distinguish between synaptic strength and morphological features. As a result, the relationship between connectivity strength and spine turnover in the hippocampus remains largely unexplored.

In this study, we used chronic two-photon calcium imaging of dendritic spines in the ipsilateral CA1 (ilCA1) region of awake mice

[1]Research Group Synaptic Wiring and Information Processing, Center for Molecular Neurobiology Hamburg, University Medical Center Hamburg-Eppendorf, Hamburg, Germany. [2]Department of Neurophysiology, Medical Faculty Mannheim, MCTN, Heidelberg University, Mannheim, Germany. ✉e-mail: simon.wiegert@medma.uni-heidelberg.de

while repeatedly stimulating presynaptic contralateral CA3 (clCA3) pyramidal neurons with optogenetics. This approach enabled us to monitor localized, synaptically evoked calcium responses at individual spines and evaluate their functional and morphological stability for over two weeks.

Our findings revealed that functionally active spines, which showed synaptically evoked calcium transients tended to form stronger and more stable connections compared to spines that did not show any synaptically evoked calcium transients. These results suggest that robust synaptic connectivity is associated with enhanced spine persistence. Overall, our results indicate that spine lifetime in the hippocampus is closely linked to synaptic strength, which may be critical for maintaining long-term synaptic connectivity.

## Results

### Identification of postsynaptic CA1 neurons by optogenetic stimulation of presynaptic CA3 neurons

To identify ilCA1 spines connected to clCA3 neurons, we sparsely labeled ilCA1 pyramidal neurons in adult mice with a Cre-dependent calcium sensor, jGCaMP7b, and implanted a chronic hippocampal window. To elicit presynaptic action potentials, we densely expressed the red light-activated optogenetic actuator ChrimsonR in clCA3 pyramidal neurons followed by the implantation of an optic fiber (Figs. 1a, S1a). We estimated the light distribution in area CA3 using a Monte Carlo simulation[18] considering light scattering and absorption at

633 nm and the optical properties of the optic fiber (200 μm, 0.22 NA). Given the modeled distribution of light intensities and the spike thresholds for ChrimsonR[19], we assume that we reliably stimulated the entire clCA3, including distal and medial regions (Fig. S1b–d). Most of the distal and medial CA3 cells project to the stratum oriens[20]. Therefore, we expect a large portion of their axons to terminate in the stratum oriens of CA1 – also on the ipsilateral side, where imaging occurred. In fact, Schaffer commissural axons from the contralateral CA3 region comprise approximately 50% of the synaptic input, preferentially terminating in the stratum oriens of the ipsilateral CA1 region[21]. Therefore, by imaging spines in the stratum oriens of the ipsilateral CA1, we expect to reliably detect postsynaptic responses to optogenetic stimulation of the contralateral CA3 (Fig. S1e, f).

During two-photon spine imaging, we recorded the motion of head-fixed mice using a passive treadmill, enabling closed-loop optogenetic stimulation selectively during periods of immobility, when motion artefacts were largely absent (Fig. 1b). Accurate, continuous monitoring of locomotion via the treadmill was essential to prevent brain-motion artefacts that would lead to out-of-focus signals (Fig. S1g, h). This closed-loop approach allowed us to exclude motion-affected frames while maximizing the number of trials without motion artefacts.

Before starting the chronic spine imaging, we first identified ilCA1 neurons receiving reliable synaptic input from clCA3 neurons (Fig. 1c). Given that only a fraction of all presynaptic CA3 neurons was

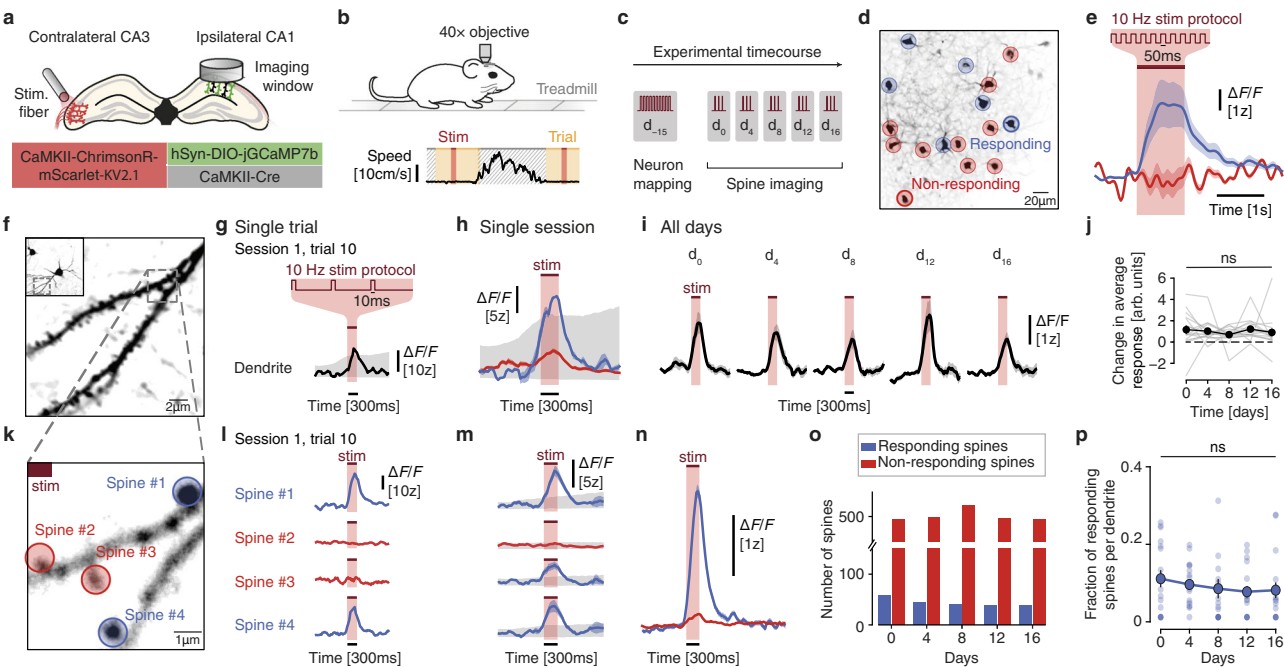

**Fig. 1 | Identification of synaptically connected CA1 neurons by optogenetic stimulation of presynaptic CA3 neurons. a** Schematic illustration depicting injections in clCA3 and ilCA1 for sparse labeling and chronic implants for stimulation and imaging. **b** Schematic illustration of closed-loop, two-photon recording configuration of a mouse on a treadmill with automated motion detection. Hashed area depicts automatically detected motion during which no trials are executed. **c** Time-course of the experiment. A strong stimulation protocol was used to identify synaptically connected neurons ('Neuron mapping'). **d** Example field of view with neurons responding (blue) or not responding (red) to optogenetic stimulation during the neuron mapping session. **e** Mean calcium transients from responding (blue) and non-responding (red) neurons shown in (**d**). Mean ± s.e.m. $N = 6$ responding neurons and 10 non-responding neurons. **f** Example field of view showing a dendrite from the neuron in the inset. **g** Example EPSCaT in the dendrite in a single optogenetic stimulation trial. Shaded gray indicates the result from the permutation test. **h** Average responses from successfully (blue) and non-

successfully (red) evoked EPSCaTs in the dendrite in (**f**). Mean ± s.e.m. Shaded gray indicates the result from the permutation test. **i** Average EPSCaTs measured at the same dendrite every four days. Mean ± s.e.m. **j** Changes in the average amplitudes of EPSCaTs in all dendrites (gray lines). Two-sided linear Mixed Model with dendrites as random effect. Mean ± s.e.m. $N = 17$ dendrites, 5 mice. **k** Zoomed-in view of dendritic segment shown in (**f**) during the optogenetic stimulation trial shown in (**g**). **l** Calcium recordings of four spines labeled in (**k**) during a single trial as in (**g**). Only spines #1 and #4 showed EPSCaTs. **m** Average EPSCaTs from the four spines in (**k**). Three spines were classified as responding (blue) and one as non-responding (red) during this session. Mean ± s.e.m. Shaded gray indicates the result from the permutation test. **n** Average EPSCaTs of all responding (blue) and non-responding spines (red) in one session. **o** Number of all responding (blue) and non-responding (red) spines over time. **p** Fraction of responding spines per dendrite over time. Two-sided linear Mixed Model with dendrites as random effect. Mean ± s.e.m. $N = 17$ dendrites, 5 mice. Source data are provided as a Source Data file.

stimulated with our approach (i.e., contralateral CA3 neurons) this ensured that we focus on those CA1 neurons that display a high fraction of functionally stimulated synapses, facilitating the detection of responding spines. Upon strong optogenetic stimulation in clCA3, a subset of ilCA1 neurons showed large, invariant calcium transients, while the remaining ilCA1 did not show detectable suprathreshold calcium events (Fig. 1d, e). The robust and reproducible responses in individual ilCA1 cells indicate that synaptic input from optogenetically activated clCA3 neurons was reliable, without leading to unspecific, global activation of the entire CA1 region. By injecting five adult mice with ChrimsonR in clCA3 and non-conditional jGCaMP8m in ilCA1 for dense labeling, we further confirmed that optogenetically evoked synaptic input to ilCA1 was maintained constant across imaging sessions (Fig. S2). In this way, we could detect the population response in ilCA1 without being biased towards a small number of sparsely labeled neurons. We stimulated clCA3 neurons using the same protocol and imaging timeline (Fig. 1c) and chronically recorded synaptically evoked calcium responses from the same population of ilCA1 neurons. We did not find any systematic changes in the mean amplitude over time, suggesting that overall optogenetically evoked synaptic transmission remained stable (Fig. S2c, d). Thus, our experimental protocol enabled us to optogenetically evoke stable synaptic transmission at Schaffer commissural synapses in awake mice during a time period of more than two weeks.

## Local spine calcium responses are evoked by subthreshold optogenetic stimulation of presynaptic neurons

Since we aimed to investigate structural plasticity in functionally connected spines, we optogenetically stimulated CA3 neurons to evoke excitatory postsynaptic calcium transients (EPSCaTs) in postsynaptic CA1 spines. EPSCaTs depend on depolarization by AMPA receptors[22] and they are largely mediated by NMDA receptors[23] with contributions from voltage-gated calcium channels[24] and intracellular stores in some cases[25]. EPSCaTs are indicative of synaptic strength[26,27] and they were recently shown to reliably report glutamatergic synaptic transmission at CA1 pyramidal neurons in vivo[28]. Thus, EPSCaTs serve as a good proxy for postsynaptic responses at spines of CA1 pyramidal neurons.

To ensure that we can detect EPSCaTs reliably, we aimed to avoid triggering strong global calcium transients in the entire ilCA1 neuron, which could arise from suprathreshold depolarization and action potential back propagation. The protocol was set to an intensity, such that most trials remained sub-threshold, triggering dendritic events only in a small number of trials ($5.90 \pm 0.90\%$) (Fig. 1f–h). Keeping the stimulation protocol constant, we then chronically recorded EPSCaTs in the same spines every four days over the course of 16 days (see timeline in Fig. 1c). To verify the reliability of our adjusted optogenetic stimulation paradigm, we first measured the average calcium transients over all trials across the entire dendrite, including suprathreshold events and all responding spines. Average calcium transients were similar across days (Fig. 1i, j), validating the stability and consistency of optogenetically evoked synaptic transmission in ilCA1.

To evaluate the individual synaptically evoked calcium events for a given spine, the contribution of dendritic events was subtracted[29,30]. This method was effective in eliminating occasional weak dendritic calcium events that would lead to false-positive EPSCaTs in spines that did not receive synaptic input. We thus could identify ilCA1 spines that received inputs from optogenetically stimulated clCA3 neurons (Figs. 1k, l, S1f). As expected from the low release probability of Schaffer commissural synapses, spines were not showing EPSCaTs at every trial. By averaging all trials of the entire session, we could identify ilCA1 spines receiving functional input from optogenetically stimulated clCA3 cells and discriminate them from spines not responding in that session (Fig. 1m, n).

When systematically following the same spines over time (every four days), we observed that most spines responded only in one session and remained silent in preceding or following sessions. (Fig. S3a, b). A small fraction of spines showed responses in two (12.0%), three (4.6%) or four (1.1%) sessions (Fig. S3b). A majority of spines showed a first response in the first imaging session (approx. 1/3 of all spines ever showing a response). Accordingly, spines showing EPSCaTs for the first time were also found in later sessions, with an equal distribution across those sessions (Fig. S3c). The bias towards the first session is inherent to the experimental approach: by default, any responding spine in this session is classified as responding for the first time, since no preceding sessions exist. In addition, since we screened for dendrites with active spines in the first session, this may also have contributed to a higher fraction. In the following sessions, the same dendrites were revisited irrespective of spine responses. Nevertheless, the variability in responses (i.e., spines not responding in all sessions or even functionally emerging in later session) suggests that functional connectivity gets reconfigured between sessions and therefore, individual synaptic inputs drift over time.

Given the observed drift at the single-synapse level, we investigated whether the total fraction of responding spines per session remained stable across days, which would ensure the preservation of average synaptic input, as suggested by the overall stability of dendritic responses (Fig. 1j). Indeed, both the total number (Fig. 1o) and relative dendritic fraction (Fig. 1p) of responding spines remained consistent across sessions, indicating stable dendritic input. This finding is further supported by the invariant spine density across all sessions (Fig. S3d).

Together, these results demonstrate that, while individual synaptic inputs from clCA3 neurons exhibited high temporal variability, the overall dendritic input remained stable throughout the experiment.

## Responsive spines are strongly connected to presynaptic partners

Next, we grouped all spines into two categories: spines that exhibited optogenetically evoked postsynaptic calcium responses at least in one session ("responsive spines") and spines that were classified as non-responding across all sessions ("unresponsive spines", Fig. 2a, see Methods for details).

We then asked whether responsive spines constituted stronger synapses than unresponsive spines. The latter group may include either weakly connected synapses of optogenetically stimulaed CA3 neurons or synapses of CA3 neurons that were not stimulated (e.g., ipsilateral CA3 neurons). Because spine volume is widely considered a proxy for synaptic strength[7,31,32], we estimated the head volumes of both responsive and unresponsive spines (Fig. 2b). Responsive spines were significantly larger than unresponsive spines (Fig. 2c, Fig. S4), suggesting that, on average, they were more strongly connected to their presynaptic partners.

However, because unresponsive spines may also include strong synapses that were not activated, we next restricted our analysis to responsive spines to avoid confounds associated with synapses of unknown activation status. We tested whether spine volume correlated with the amplitude of excitatory postsynaptic calcium transients (EPSCaTs). Indeed, we observed a positive correlation between spine volume and EPSCaT amplitude (Fig. 2d), consistent with previous findings that larger spines correspond to stronger synapses[33]. Thus, although unresponsive spines likely represent a heterogeneous population with a broad distribution of synaptic strengths, our results indicate that responsive spines preferentially correspond to strong synapses.

To ensure that our spine volume estimates were not influenced by intracellular calcium levels, we acquired volumetric stacks at an isosbestic (calcium independent) excitation wavelength (Fig. S5a). We

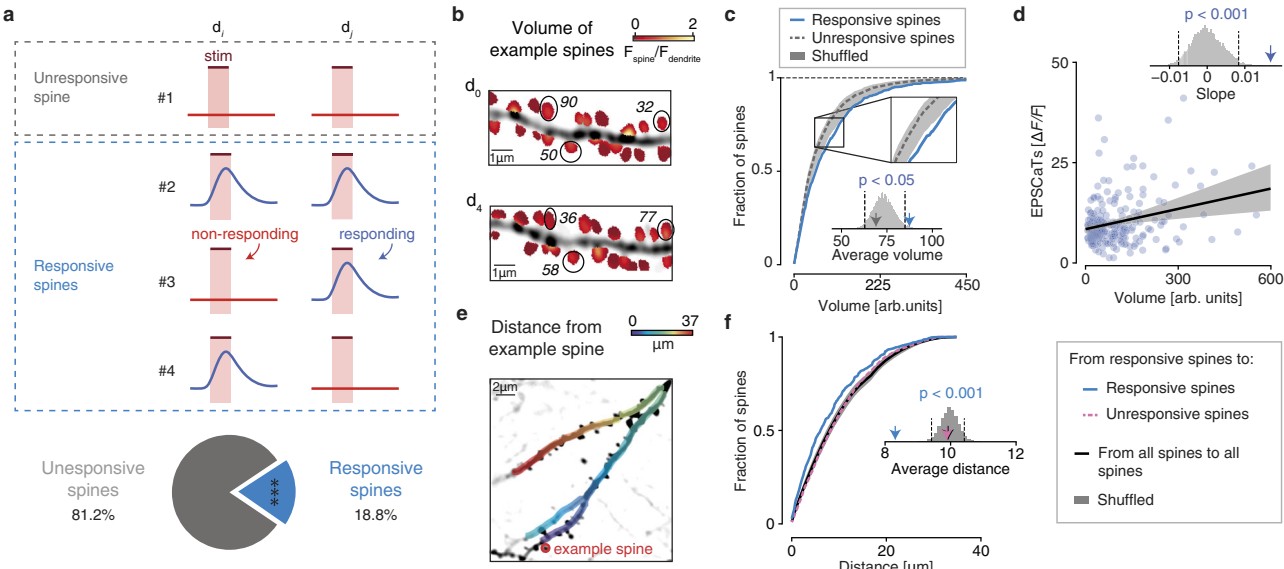

**Fig. 2 | Responsive spines display strong synaptic connections. a** Definition of responsive and unresponsive spines and their fractions in the total pool of analyzed spines. Top: an unresponsive spine was classified as non-responding in all sessions (#1), while responsive spines showed EPSCaTs in at least one imaging session (#2-4). Bottom: total fraction of responsive spines. $N = 175$ responsive spines and 754 unresponsive spines, 17 dendrites, 5 mice. Two-sided binomial test. ***$p < 0.001$. **b** Example of relative spine volumes on a dendrite at two different timepoints. The relative volume was derived by normalizing total spine fluorescence to dendrite fluorescence (see Methods). **c** Cumulative distributions of spine head volumes of responsive (blue) and unresponsive (dark gray) spines. Inset: Shuffled volumes indicated by gray histogram. Arrows indicate average spine volumes. Two-sided permutation tests. $N = 175$ responsive spines and 754 unresponsive spines, 17 dendrites, 5 mice. $p = 0.01$. **d** EPSCaTs amplitude as a function of the volume of

responsive spines. Linear regression indicated by black line and shuffled data indicated by shaded gray area. Top: arrow indicates slope and gray histogram shows shuffled relationship between EPSCaTs and volume. Pearson correlation: 0.27. Two-sided permutation tests. $N = 175$ responsive spines, 17 dendrites, 5 mice. $p = 3.9 \times 10^{-5}$. **e** Dendritic path length (distance) between a given spine (example spine) and all other spines. The color code indicates the distance between the example spine and all other spines present along the dendritic segment. **f** Cumulative distributions of distances between spines. Inset: Shuffled distances indicated by gray histogram. Arrows indicate average differences between spine categories. Two-sided permutation tests. $N = 175$ responsive spines and 754 unresponsive spines, 17 dendrites, 5 mice. $p = 0.0001$ (responsive), 0.74 (unresponsive). Source data are provided as a Source Data file.

observed a strong correlation between the two measures (Fig. S5b), indicating that both strategies yielded comparable volume estimates. In addition, we also controlled for the relationship between the measured spine volume and calcium fluctuations (Fig. S5c). We did not find a significant correlation, suggesting that volume measurements were not confounded by dynamic brightness changes arising from intracellular calcium dynamics. Finally, we asked whether spine volume was directly related to the presence of calcium responses within individual imaging sessions. To address this, we subdivided all responsive spines into sessions in which a calcium response was detected and sessions in which no response was observed, and compared the corresponding spine volumes. We found no significant effect of response presence on measured spine volume, although we observed a weak trend toward slightly larger volumes during responding sessions (Fig. S5d). Thus, while spine volume correlated with EPSCaT amplitude across spines, this relationship was only weakly expressed within individual spines across sessions, likely reflecting a non-linear relationship between calcium transients, synaptic strength, and spine morphology.

Next, we investigated the dendritic distribution of responsive spines. Previous studies suggested that synaptic clustering can enhance local dendritic integration, potentially leading to cooperative synaptic potentiation[20,22,23]. Moreover, due to their highly divergent connectivity, CA3 neurons form only a small number of synapses with a given CA1 neuron. These synapses are often found in close proximity to each other, given that they arise from the same presynaptic axon branch[34]. To assess whether responsive spines were in closer proximity to each other compared to all other spines, we measured all-to-all dendritic distance between the spines in the field of view (Fig. 2e). Our comparison of the spatial distribution of responsive and all other spines revealed a closer spatial proximity (shorter dendritic length)

between responsive spines versus unresponsive spines (Fig. 2f). This suggests – as expected from CA3-CA1 connectivity – that functional inputs from a defined population of presynaptic neurons are more likely to be in spatially closer proximity rather than randomly distributed[35].

## Responsive spines display increased stability

Since responsive spines were on average larger than randomly selected unresponsive spines, and given the positive correlation between spine volume and postsynaptic responses, we asked whether they also exhibited greater stability over time. Spines were categorized into four dynamic categories: (1) persistent spines, present throughout the entire time series; (2) transient spines, which appeared temporarily and then disappeared again; (3) formed spines, which emerged during the time series and remained until the end; and (4) eliminated spines, present at the beginning but disappearing before the end of the time series (Fig. 3a).

Independently, we categorized spines as either responsive or unresponsive and analyzed their distribution across the four groups. We found that responsive spines exhibited a higher proportion in the persistent category and a lower proportion in the transient category compared to unresponsive spines (Fig. 3b). Transient spines in the responsive group were more likely to reappear at the same dendritic location on different days (i.e., they were recurrent) compared to transient unresponsive spines (Fig. 3c). Despite this, the frequency of flips (disappearance and reappearance) was similar between transient responsive and unresponsive spines. This recurrence in responsive spines may be explained by repeated stimulation, which could act as a stabilizing signal for the synapse[36,37], promoting the regrowth of otherwise retracting spines.

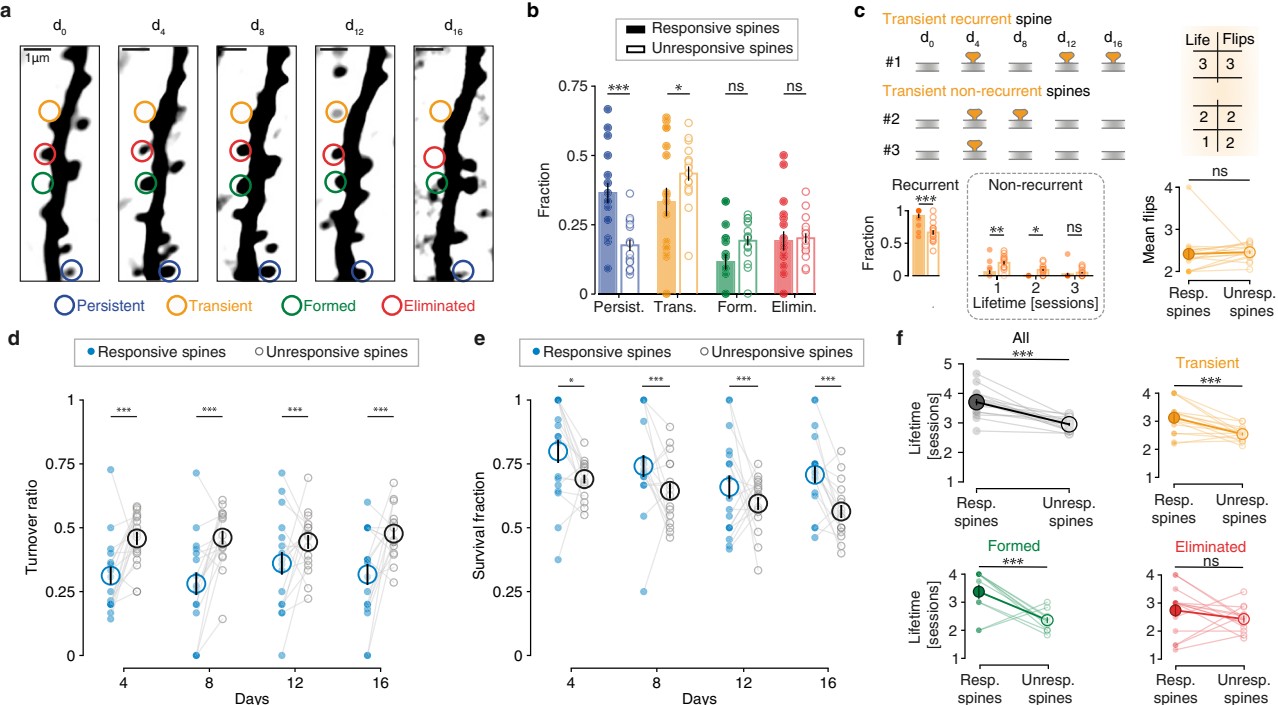

**Fig. 3 | Enhanced stability of responsive spines. a** Example dendrite imaged every four days. Examples of persistent (blue), transient (yellow), formed (green), and eliminated (red), spines are indicated by colored circles. **b** Fractions of the different categories for responsive (filled circles) and unresponsive spines (empty circles). Two-sided linear Mixed Model with dendrites as random effect and pairwise comparisons with Bonferroni correction. Each circle represents a dendrite. Mean ± s.e.m. $N = 17$ dendrites, 5 mice. $p = 0.0001$ (persistent), 0.02 (transient), 0.09 (formed), 0.83 (eliminated). **c** Characteristics of transient spines. "Life": Cumulative lifetime in days. "Flips": Number of appearance and disappearance events. Two-sided linear Mixed Model with dendrites as random effect and pairwise comparisons with Bonferroni correction. Mean ± s.e.m. $N = 17$ dendrites, 5 mice. Recurrent: $p = 0.0001$; non-recurrent: $p = 0.0001$ (1 session), 0.02 (2 sessions), 0.48 (3 sessions); mean flips: $p = 0.72$. **d** Turnover ratio of responsive spines versus unresponsive spines in a dendritic segment between two successive sessions. Pairs of filled and open circles connected by a line represent responsive and

unresponsive spines on a given dendrite. Two-sided linear Mixed Model with dendrites as random effect and pairwise comparisons with Bonferroni correction. Mean ± s.e.m. $N = 17$ dendrites, 5 mice. $p = 0.0001$ (4 days), 0.0001 (8 days), 0.0001 (12 days), 0.0004 (16 days). **e** Survival fraction of responsive spines versus unresponsive spines at each session. Pairs of filled and open circles connected by a line represent responsive and unresponsive spines on a given dendrite. Two-sided linear Mixed Model with dendrites as random effect and pairwise comparisons with Bonferroni correction. Large circles: mean ± s.e.m. $N = 17$ dendrites, 5 mice. $p = 0.02$ (4 days), 0.0001 (8 days), 0.0001 (12 days), 0.0001 (16 days). **f** Lifetime of responsive (filled circles) and unresponsive spines (empty circles) sorted by category. Categories colored as in (**b**). Two-sided linear Mixed Model with dendrites as random effect. $N = 17$ dendrites, 5 mice. $p = 2.7 \times 10^{-7}$ (all), 0.0005 (transient), 0.0004 (formed), 0.24 (eliminated). ns: non-significant, *$p < 0.05$, **$p < 0.01$, ***$p < 0.001$. Source data are provided as a Source Data file.

To further characterize the dynamics of responsive and unresponsive spines, we analyzed different properties related to their life expectancy in more detail. First, we separately assessed the turnover ratio for responsive and unresponsive spines in each imaged dendritic segment. For both groups (responsive or unresponsive spines), the turnover ratio is defined as the sum of spine formation and spine elimination rates between consecutive imaging sessions. Consistent with the higher fraction of persistent spines in the responsive population, the turnover ratio was lower for responsive spines at each imaging interval (Fig. 3d). Since the turnover ratio depends on both formation and elimination rates, we analyzed both parameters separately. We found that both the formation and elimination rates of responsive spines were lower compared to unresponsive spines (Fig. S6), as expected, given the larger population of stable responsive spines.

The turnover ratio, including formation and elimination rates when analyzed individually, remained invariant within both populations (i.e., responsive and unresponsive spines), indicating that optogenetic stimulation did not directly influence spine stability. This conclusion was further supported by comparing the stability of all spines in the group of optogenetically stimulated animals to spines on dendrites from control animals not undergoing optogenetic stimulation (Fig. S7). We did not observe any differences in the distribution of

the dynamic categories, persistent, transient, formed, eliminated nor in the turnover ratio, or survival fraction between the two treatment conditions (Fig. S7a–c). Thus, similar to optogenetically stimulated dendrites (Fig. S3d), the spine density on control dendrites remained stable over days (Fig. S7d). Finally, the volume of non-stimulated control spines was not significantly different from responsive and unresponsive spines in optogenetically stimulated mice (Fig. S7e). Together with the stable amplitude of EPSCaTs in dendrites (Fig. 1i), these results confirm that the optogenetic stimulation used to identify synaptic connections did not induce significant changes in spine stability or structure over time.

A high turnover ratio could potentially result from a high formation rate of new spines without affecting the stability of existing spines. To explore this in more detail, we assessed the survival fraction defined as those spines detected in a given imaging session that were present in all preceding imaging sessions from the beginning. The survival fraction of responsive spines was consistently higher than that of unresponsive spines, further highlighting their greater stability (Fig. 3e). The increased stability and extended lifetime of responsive spines (Fig. 3f) may largely be attributed to the higher proportion of persistent spines in this population. To investigate whether spine lifetime is generally higher when spines are responsive, we specifically measured the lifetimes of all non-persistent spines (i.e., transient,

formed, and eliminated spines). Remarkably, spines that formed during the experiment (transient and formed spines) exhibited longer average lifetimes when they were responsive to presynaptic optogenetic stimulation (Fig. 3f). These findings suggest that responsive spines, which tended to be stronger than unresponsive spines, also exhibited a longer overall lifetime.

To validate this relationship independently of optogenetically identified spines, we categorized all spines—regardless of their activity—again into the four dynamics categories persistent, transient, formed, and eliminated, hypothesizing that persistent spines would exhibit larger volumes, indirectly suggesting that they are stronger than non-persistent ones. 19.7% of all spines were classified as persistent, while nearly half were transient (45.6%). The remaining spines were divided almost equally between formed (16.9%) and eliminated (17.8%) categories (Fig. 4a). As expected from the responding spine population and in agreement with previous work[32], persistent spines were significantly larger than spines in all other non-persistent categories (Fig. 4b, Fig. S8).

We also analyzed the spatial distribution of the four spine categories and found a shorter distance between persistent spines compared to their distance to other spines (Fig. 4c, Fig. S9), suggesting that persistent spines tended to be closer together. In comparison, no difference to the overall median pairwise distance was found in the distance between transient, eliminated or formed spines. Notably, newly formed and transient spines were more distal to persistent spines. Instead, newly formed spines appeared closer to the location of eliminated spines, suggesting that they appeared in places where preexisting spines were lost. Conversely, transient spines were distant to eliminated spines.

In conclusion, persistent spines showed higher proximity to each other and were generally larger than non-persistent spines of all categories – suggesting a stronger connection to their presynaptic partners, while non-persistent spines were likely less strongly connected and more randomly distributed. This analysis further corroborates the positive relationship between synaptic strength, size, and stability.

We have established a relationship between synaptic strength (EPSCaT amplitude) and spine volume in vivo (Fig. 2e). Moreover, we showed that spines displaying a postsynaptic response to presynaptic optogenetic stimulation are more persistent than unresponsive spines (Fig. 3) and that stable spines display a larger head volume (Fig. 4), indirectly suggesting that larger spines belong to stronger synapses, which, in turn, display higher stability. To establish this relationship between synaptic strength, spine volume and spine stability more firmly, we asked if the frequency of synaptic responses (i.e., the number of imaging sessions in which spines showed EPSCaTs) is related to EPSCaT amplitude and spine head size. We found a significant positive correlation between the frequency of responses and the average volume of responding spines as well as EPSCaT amplitude (Fig. 4d, e).

Because our categorization into responsive and unresponsive spines could have led to the exclusion of weakly responsive spines, we considered again all imaged spines and split them into five categories according to their highest maximum fluorescent transients across all sessions, including all spines that were previously categorized as unresponsive (Fig. 4f). This alternative analysis confirmed the relationship between the maximal calcium response and the spine lifetime (Fig. 4g). Thus, our data demonstrate a close relationship between spine size, functional connectivity and spine stability at hippocampal CA3-CA1 synapses in vivo.

## Discussion

In this study, we demonstrated that the stability of glutamatergic spines at CA3-CA1 synapses in the hippocampus is regulated by connectivity strength. Using chronic two-photon imaging of spines in ilCA1, combined with optogenetic stimulation of presynaptic clCA3

neurons, we discovered that spines that exhibited robust postsynaptic calcium responses displayed increased size, a longer lifetime, and a closer spatial proximity to each other compared to neighboring spines that did not exhibit detectable postsynaptic calcium responses.

Previous studies in anesthetized mice have reported a wide range of spine dynamics at CA1 basal dendrites in vivo[14–16]. However, these studies have only assessed anatomical features without linking spine turnover with their functional properties. Therefore, it remained unclear to what extent functional and morphological properties, such as synaptic weight or spine volume, would predict spine stability at hippocampal synapses in vivo. By tracking functionally identified spines over more than two weeks in awake mice, we established a link between spine lifetime and connectivity strength, while avoiding the confounding effects of repeated anesthesia[15]. Spines with a strong functional connection (i.e., responsive spines) not only exhibited a larger head volume but also showed a lower turnover ratio, which was due to an increased survival rate and thus, an enhanced lifetime compared to their unresponsive neighbors. This work is in line with a previous study, showing a correlation between spine lifetime and spine size[16]. In addition to the increased stability of responsive spines we also found a correlation between the propensity of responses and spine volume, indicating that the stability of functional connectivity directly influences the weight and thereby, the persistence of a synapse, as previously suggested by anatomical studies[13,38,39].

The larger head volume of responsive spines compared to unresponsive neighboring spines indicates a stronger synaptic connection[7,31,32] with presynaptic clCA3 neurons. Those spines could belong to functional ensembles, which previously underwent LTP[40,41] as suggested by a previous report[35]. We observed a slightly higher spatial proximity between strong, responsive synapses, which may indicate that small groups of spines receive input from the same presynaptic CA3 cells or from CA3 cells with similar functional tuning. These strong inputs may have contributed to postsynaptic calcium responses in CA1 neurons during our initial screening (Fig. 1c–e) by contributing sufficient depolarization to reach cellular spike threshold[41]. Notably, the proximity of responsive spines in our study does not represent anatomical clusters. This is likely due to the fact that we stimulated only a subset of all CA3 cells providing direct synaptic input to the postsynaptic CA1 cell. Thus, only a fraction of all spines on a given dendritic segment receives direct, optogenetically evoked input, while other synapses on the same dendrite may originate from non-stimulated CA3 cells, which remain elusive during postsynaptic calcium imaging. Functional clusters might only be revealed by activity of all presynaptic neurons that share functional properties, as is the case during spatial navigation where spatially tuned CA3 cells are co-active in a given dendritic location[42,43]. Nonetheless, the closer spatial proximity of responsive spines aligns with findings from those recent studies demonstrating that spine clusters on place cells were correlated with the spatial tuning of these neurons during a spatial navigation task[44]. As a consequence, such clustering of functionally related spines likely contributes to neuronal firing within specific place fields, consistent with prior studies that have linked functional spine clustering to learning[31,42,43,45]. Due to their increased stability, such synaptic populations can form long-lasting ensembles with a feature-specific presynaptic population of CA3 cells. This helps to stabilize specific synaptic input to a feature-selective CA1 cell (e.g., place cell firing) over time despite fluctuations at individual synapses.

Our stimulation paradigm induced local synaptic EPSCaTs in postsynaptic spines of CA1 neurons. A potential limitation of this method is its reliance on voltage-gated calcium entry[23–25,27,46] which only indirectly reports glutamatergic synaptic transmission. Thus, weak synapses or spines with low AMPA receptor density may not have reached sufficient depolarization to unblock NMDA receptors or open voltage-gated calcium channels. Therefore, weakly connected synapses might have fallen under the detection threshold and therefore

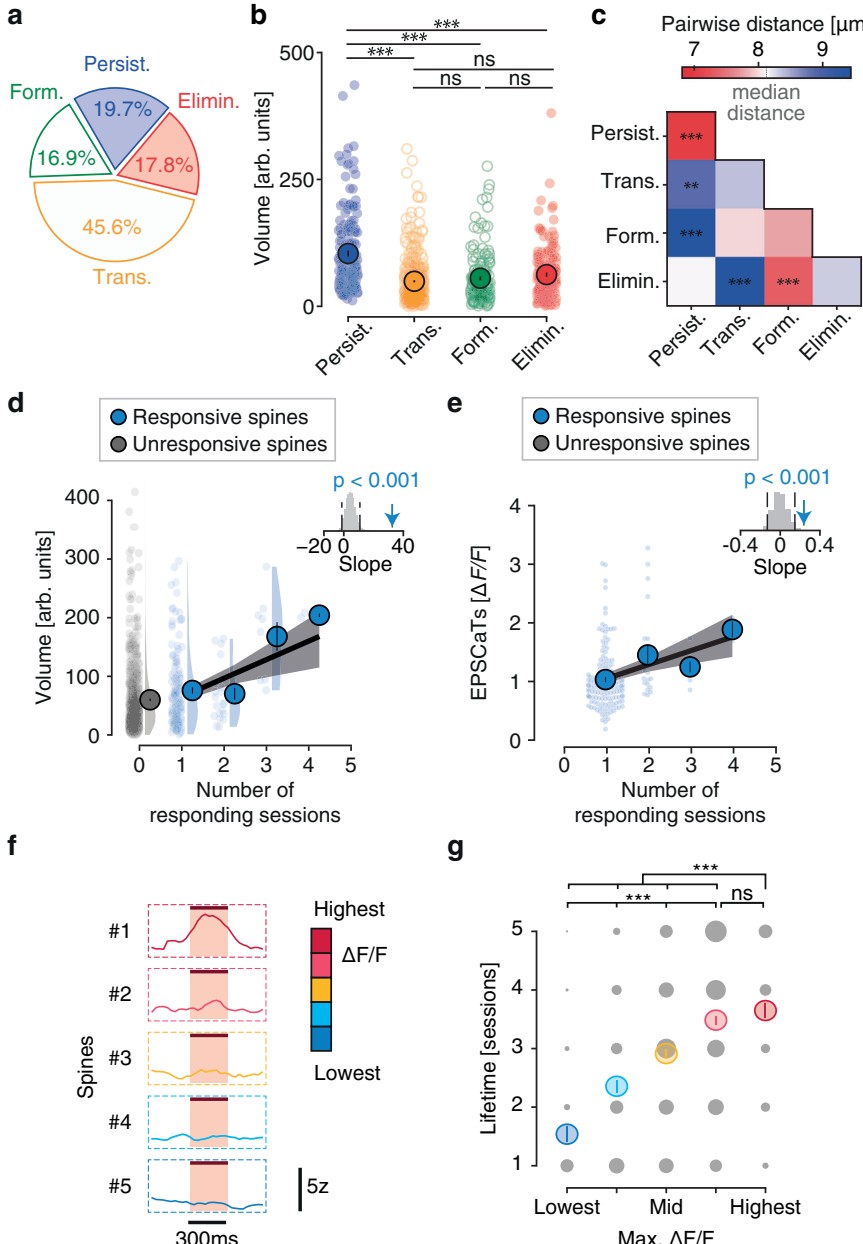

**Fig. 4 | Persistent spines are larger, more proximal to each other, and show larger EPSCaTs compared to non-persistent spines. a** Percentages of the different dynamic categories of all imaged spines. $N = 929$ spines, 17 dendrites, 5 mice. **b** Head volumes of the different dynamic spine categories. $N = 929$ spines, 17 dendrites, 5 mice. Two-sided permutation tests. $P = 0.0001$ (persistent vs. transient), 0.0001 (persistent vs. formed), 0.0001 (persistent vs. eliminated), 0.29 (transient vs. formed), 0.27 (transient vs. eliminated), 1.00 (formed vs. eliminated). **c** Pairwise distances between spines of the different dynamic spine categories. $N = 929$ spines, 17 dendrites, 5 mice. Two-sided permutation tests. Spine head volume (**d**) or EPSCaT amplitude (**e**) as a function of the number of sessions in which the spine responded. Mean ± s.e.m. Linear regression indicated by black line and shuffled data indicated by shaded gray area. Top right: arrow indicates slope

and gray histogram shows shuffled relationship between volume (**d**) or EPSCaT amplitude (**e**) and the number of responding sessions. Two-sided permutation tests. $N = 175$ responsive spines, 17 dendrites, 5 mice. $p = 0.0001$ (volume), 0.0009 (EPSCaTs). **f** Example traces of the spines sorted according to their maximal ΔF/F irrespective of their categorization as responsive or unresponsive. **g** Lifetime of the spines categorized according to their maximal ΔF/F. $N = 929$ spines, 17 dendrites, 5 mice. Two-sided permutation tests. $p = 0.0008$ (lowest vs. low), 0.0001 (lowest vs. mid), 0.0001 (lowest vs. high), 0.0001 (lowest vs. highest), 0.0003 (low vs. mid), 0.0001 (low vs. high), 0.0001 (low vs. highest), 0.0001 (mid vs. high), 0.0001 (mid vs. highest), 0.78 (high vs. highest). ***$p < 0.001$, **$p < 0.01$, ns: non-significant. Source data are provided as a Source Data file.

gotten classified as unresponsive spines. Consequently, the absence of spine calcium responses can have two reasons. Either the spine belonged to a weak, stimulated synapse, which did not elicit suprathreshold postsynaptic calcium transients due to insufficient depolarization, or the spine received input from a non-stimulated CA3 cell. Thus, unresponsive spines likely represent a mixed population of synapses of various strengths. Importantly, a recent study in awake

mice showed that EPSCaTs in CA1 spines reliably reported glutamatergic transmission, suggesting that EPSCaTs are suitable to monitor excitatory transmission at Schaffer collateral/commissural synapses. Moreover, since EPSCaTs depend on depolarization by AMPARs[22] they are indicative of synaptic strength[26]. A relationship between synaptic strength and EPSCaTs was further demonstrated in vivo via simultaneous spine voltage imaging[47]. Therefore, EPSCaTs most likely scale

with synaptic weight[48,49]. This relationship is further underlined by the positive correlation between EPSCaT amplitude and spine volume observed in this study. Thus, synapses that showed robust EPSCaTs were considered strong synapses, while synapses that did not exhibit EPSCaTs were either weak or they belonged to a pool of non-stimulated synapses with undetermined strength. To rule out any potential confounds associated with the classification into responsive and unresponsive spines we additionally sorted all synapses according to their maximal fluorescence transients and analyzed their lifetime, which confirmed that spines with large calcium responses had a higher survival rate (Fig. 4f).

Overall, 19% of all recorded spines displayed robust calcium responses. Notably, responsive spines exhibited EPSCaTs primarily in a single imaging session, suggesting dynamic functional reconfiguration of connectivity across sessions. This reconfiguration may reflect changes in synaptic connectivity strength, wherein both spontaneous fluctuations in presynaptic AP-evoked transmitter release[50] and fluctuations in postsynaptic calcium responses could contribute to the transient loss of detectable EPSCaTs at weaker synapses. Future studies aiming to directly study the influence of synaptic plasticity on synapse dynamics and spine stability in the hippocampus, similar to prior in vitro research[48,49,51] need to overcome challenges associated with classical plasticity-induction protocols[49,52] such as high-frequency or burst stimulation in vivo. In our hands, such protocols strongly altered network dynamics and led to perception of this overall altered network state by the animal. This made it impossible to dissociate pathway-specific plasticity from global alterations of network states. Refined and spatially restricted stimulation protocols with better temporal control, for example, via holographic 2P-stimulation may help to overcome such limitations in the future and help to better understand the influence of long-term potentiation and depression on spine stability in vivo. Another strategy to circumvent artefacts associated with plasticity-inducing stimulation protocols is to rely on sensory stimuli[53] or to pair naturally occurring synaptic input with postsynaptic manipulation of neuronal activity[47,54]. Recent studies have employed postsynaptic optogenetic stimulation of CA1 cells to induce behavioral time scale plasticity followed by functional measurements of naturally occurring synaptic transmission[47,55]. While these studies linked synaptic plasticity to cellular function in vivo, they did not follow the functional or structural properties of these synapses over time. Thus, future functional and structural studies of identified spines associated with a specific behavior or a cellular function over extended time periods will be important to shed light on the role of individual synapses in long-term information processing and storage in the brain. Indirect evidence suggests that spine stability is increased at CA1 neurons, which are part of an engram[56]. Directly assessing the lifetime of spines involved in the control of these neurons and concurrent animal behavior would provide critical insights into the synaptic mechanisms underpinning memory encoding and storage within the neuronal networks involved in episodic memory formation and storage[57].

## Methods

### Animals
Adult (2 to 9 months of age) C57BL/6 J mice of both sexes were used in this study. The mice originated from the Charles River company and were housed under pathogen-free conditions at the University Medical Center Hamburg-Eppendorf with an inverted light/dark cycle of 12/12 h. The humidity and temperature in the room were kept constant (40% relative humidity; 20 °C). Food and water were available ad libitum. All procedures were performed in compliance with German law according and the guidelines of Directive 2010/63/EU. Protocols were approved by the Behörde für Gesundheit und Verbraucherschutz of the City of Hamburg under the license numbers 32/17 and 33/19.

### Mouse surgery and virus injections
C57BL/6 J wild-type mice were anesthetized with 2% isoflurane/1 L of $O_2$. The fur on the head was trimmed and carefully removed to avoid later contamination in the surgical field. The skin was disinfected using Betaisodona and mice were transferred to the stereotaxic frame. Anesthesia was maintained with 1.5% isoflurane/1 l of $O_2$. To evaluate the depth of anesthesia and analgesia, the paw withdrawal reflex test was performed with a toe-pinch. Upon deep anesthesia, Buprenorphine (0.1 mg/kg) and Carprofen (4 mg/kg) were injected subcutaneously. Mice were kept on a heating pad throughout surgery to maintain the body temperature and eye ointment (Vidisic, Bausch + Lomb) was used to prevent eye drying. A 3–4 cm midline scalp incision was made close to the injection sites. The skin was pushed to the side and a bone scarper (Fine Science Tools) was used for cleaning of the skull bone. Above the injection sites, two holes were made using a dental drill (Foredom). 0.3 μL of AAV9-CaMKII-ChrimsonR-mScarlet-KV2.1 ($1 \times 10^{13}$ vg/mL, AddGene # 124651-AAV9) viral suspension was first injected in left CA3 (−2.0 mm AP, −2.3 mm ML, −2.5 mm DV relative to Bregma) using a custom-made air-pressure driven injection system. For the spine recording experiments, 0.5 μL of a 1:1 mix of AAV9-CaMKII-Cre ($3 \times 10^9$ vg/mL, gift from D. Kuhl) and AAV1-Syn-flex-jGCaMP7b-WPRE ($1.4 \times 10^{13}$ vg/mL, AddGene #104493-AAV1) or AAV9-hsyn-DIO-GCaMP7b-P2A-GSG-mRuby3 ($8.15 \times 10^{13}$ vg/mL, AddGene #130900) was injected in right CA1 (−2.0 mm AP, +1.5 mm ML, −1.5 mm DV relative to Bregma). In experiments where dense CA1 cellular recordings were done, mice were instead injected with 0.5 μL of AAV9-Syn-jGCaMP8m ($4 \times 10^{12}$ vg/mL, AddGene #162375-AAV9) in CA1. The scalp was sutured after injections were completed. Mice were removed from anesthesia and allowed to recover in a clean cage on a heating blanket. During the following three days, mice were provided with Meloxicam mixed into soft food.

After a recovery period of at least two weeks, implantation of a hippocampal imaging window together with a contralateral optic fiber took place. Mice were anesthetized as described above. After the removal of fur, the skin covering the implantation sites was removed. The skull was cleaned and roughened with a bone scraper. A first hole using the dental drill was made over the left CA3 (−2.0 mm AP, −3.5 mm ML, tilted 35° towards the left) and an optic fiber (200 μm, NA 0.22, 1.25 mm ferrule diameter, 1.3 ± 0.1 mm length, Doric Lenses) was inserted and glued to the skull. Once the glue had dried, a circular 3 mm bone piece, centered on the CA1 injection site, was carefully removed using a trephine (MW Dental, ISO 020). The dura and somatosensory cortex above the hippocampus were carefully aspirated until the fiber tracts of the corpus callosum became visible. Sterile PBS was used to wash the craniotomy all along, and a custom-made hippocampal imaging window was inserted. To build the window, a hollow glass cylinder was glued to a No. 1 coverslip on the bottom with UV-curable glass glue (Norland NOA61). The imaging window and a head plate (Luigs & Neumann) were attached to the skull with cyanoacrylate gel (UHU SuperGel). Dental cement (Super Bond C&B, Sun Medical) was then applied until the complete closure of the cranial surgery. As before, animals were provided with care after the end of the surgery and could recover for at least two weeks before the beginning of experiments.

### Two-photon imaging
Mice were progressively handled and habituated to the imaging setup and head fixation for at least one week before starting imaging experiments. Additional habituation sessions were occasionally added until the mice showed no more signs of stress, usually expressed as uninterrupted and disorganized runs. Mice were placed on a 2m-long linear treadmill (Luigs and Neumann) for close-loop experiments described below. Using a treadmill, we could better monitor motion that can lead to out-of-focus signal, discarding those frames for analysis while maximizing the number of trials without motion artefacts.

**Table 1 | Duration of imaging sessions**

|        | Mouse #1 | Mouse #2 | Mouse #3 | Mouse #4 | Mouse #5 |
|--------|----------|----------|----------|----------|----------|
| Day 0  | 133 min  | 70 min   | 36 min   | 54 min   | 45 min   |
| Day 4  | 95 min   | 71 min   | 22 min   | 44 min   | 38 min   |
| Day 8  | 75 min   | 64 min   | 19 min   | 44 min   | 34 min   |
| Day 12 | 54 min   | 61 min   | 19 min   | 40 min   | 38 min   |
| Day 16 | 60 min   | 128 min  | 20 min   | 34 min   | 59 min   |

Moreover, a treadmill was chosen, since this provides the animals with the opportunity to voluntarily move during the imaging session and therefore is less stressful compared to full immobilization. A 633 nm laser (Coherent) was connected to the implanted optic fiber on the mouse head through two patch cords. From the laser combiner (LightHUB, Omicron), a main patch cord (Doric Lenses, optic fiber 200 µm, NA 0.22, 2 m long, SMA-SMA) was connected to a patch cord (Doric Lenses, optic fiber 200 µm, NA 0.22, 1 m long, SMA-MF1.25), which itself was connected to the implanted optic fiber through a dark mating sleeve (Doric Lenses). To avoid light contamination from the optogenetic stimulation, a notch filter (Semrock, #SP01- 633RU-32) was inserted before the photomultipliers, and the path between the objective and the photomultipliers was further covered with black tissue. The implanted chronic window was centered under the microscope using widefield epifluorescence before starting two-photon imaging. jGCaMP7b was excited with a Ti:Sa laser (Chameleon Vision-S, Coherent) tuned to 930 nm at 10 to 40 mW through a 40× water immersion objective (Nikon CFI 40×, 0.80 NA, 3.5 mm WD). Single planes (512 × 512 pixels) were acquired at 30 Hz with a resonant scanner using ScanImage 2017b (in MATLAB 2017b). In the first session, multiple fields of view (FOV) per mouse were acquired. For each FOV, optogenetic stimulation using a strong protocol (20 trials, 10 Hz, 10 pulses of 50 ms each, spaced by 5 s) was used to evoke neuronal postsynaptic calcium transients. A custom Python script was used to control the laser, to trigger the beginning of the session, and to trigger the onset of each trial. The treadmill was continuously monitored to initiate optogenetic stimulation trials only when the mouse was quiet for at least one second before trial onset. After post-hoc identification of responding neurons as described below, imaging of dendrites of responding neurons was performed 15 days later in consecutive imaging sessions every four days. For this, segments horizontal to the imaging plane on first or second order basal dendrites in close proximity of the soma were chosen. The global signal-to-noise ratio (SNR) from each dendritic segment was calculated as follows: (1) the signal amplitude was estimated by taking the 95th percentile minus the 5th percentile of the dendritic trace, (2) the noise was estimated by calculating the standard deviation of the high-pass filtered dendritic trace (cutoff: 10 Hz), (3) the signal amplitude was divided by the noise estimation. Dendritic segments with a signal-to-noise ratio below 8 for any of the recorded sessions were discarded (average global dendritic SNR: 16.2 and range: 8–44, average global spine SNR: 7.5 and range: 3–32). For each session and each dendrite, subthreshold optogenetic stimulation was performed (40 trials of 3 pulses of 10 ms at 10 Hz, spaced by 5 s). At the beginning of each session, for each mouse, the order of imaged dendrites was shuffled to account for potential influence of the optogenetic stimulation across the session. The lengths of imaging sessions depended on the number of imaged dendrites as well as the periods of spontaneous running activity of the mouse. The session durations are listed in Table 1.

## Histology

Mice were injected with a lethal dose of ketamine and xylazine and transcardially perfused with PBS followed by 4% paraformaldehyde (PFA). Brains were removed and stored in 4% PFA. Brains were sliced

at 60 µm using a vibratome (VT1000S, Leica). Slices were mounted on microscope slides using an aqueous mounting medium (FluoroMount) and coverslips (1871, Carl Roth). Multi-tile overview images of native fluorescence of the expressed fluorophores were acquired with an epifluorescence microscope (AxioObserver, Zeiss) using the 10x objective (Plan-Apochromat, Zeiss) for slice overviews. For high-resolution images a confocal microscope (LSM 900, Zeiss) with a 20x objective (Plan-Apochromat, Zeiss) was used. For both, preset parameters for eGFP and mCherry were used and gain, light intensity and exposure time were adjusted for each sample.

## Data analysis

**Image processing.** Time series were motion-corrected using Suite2p (version 0.10.2)[58]. The shift between frames from Suite2p was further used to distinguish between periods of quiescence and locomotion. For dendritic imaging, time(t)-projections from each motion-corrected recording for each day were made. T-projections were registered using the rigid body or translation mode of the plugin StackReg[59] of the open-source platform Fiji[60]. Images were visually inspected and dendrites presenting excessive motion were removed from further analyses. Regions of interest (ROIs) were manually drawn around visually identified spines on motion-corrected frames in Fiji. ROIs were separately drawn for dendrites and the background. Calcium traces were extracted on the motion- corrected frames using a custom-made script by averaging the fluorescence of all pixels of each ROI (Fiji).

**Trial selection.** All data were analyzed using custom-made scripts with Python (Python Software Foundation) installed on an open-source Anaconda environment (Anon, 2020. *Anaconda Software Distribution*, Anaconda Inc. https://docs.anaconda.com/) and the PyCharm integrated development environment (JetBrains). First, residual motion present in motion-corrected recordings was identified using cross-correlation on the motion-corrected frames and excluded from the analysis. To be considered, trials were required to constitute of at least 1 s of stable baseline (without excessive motion as determined ahead of trial start), and at least 70% of the data points during stimulation needed to be stable.

**Fluorescence transient analysis.** In time series at the cellular resolution, the extracted fluorescence from Suite2p was used. To correct for fluorescence bleaching and to increase the signal-to-noise ratio in the data, a baseline correction was applied. For each neuron, the neuropil was subtracted from the raw fluorescence: $F_{neuron} = F_{raw} - 0.7 \times F_{neuropil}$[30,58,61]. To identify neurons responding to optogenetic stimulation, the average trace of the baseline-corrected calcium signal after stimulation was used for each neuron. A neuron was classified as responding if the fluorescence in > 50% of the time points during the optogenetic stimulation was greater than 95% of the values of the mean-shuffled ΔF/F (10,000 shuffles, $p < 0.05$).

To identify dendritic transients, the distribution of the ROIs data points for each neuron was used. Using a bootstrap analysis, the null distribution was obtained by permutation of the onset of the optogenetic stimulation at random locations. Then, significant optogenetically-evoked dendritic transients were detected as ΔF/F greater than the ΔF/F obtained from the 95% of the values of the mean-shuffled ΔF/F (10,000 shuffles, $p < 0.05$). A transient was further confirmed if, during the optogenetic stimulation, more than 30% of the data points were significant as described above.

Because suprathreshold calcium events potentially mask the spine-specific calcium transients, robust regression was used to reduce the contamination of dendritic calcium activity on spine calcium traces, as previously reported[29,30]. To decrease the high-frequency noise in traces, spine calcium traces were convolved using a 5-bin boxcar.

For each spine, an average trace using all trials of a given session was calculated to determine if the spine showed excitatory post-synaptic calcium transients (EPSCaTs). Permutations were made to find statistically significant time points during the optogenetic stimulation window. To minimize false-positive spines due to noise in the signal, as for dendrites, a spine was classified as responding if the calcium signal was greater than 95% of the values of the mean-shuffled $\Delta F/F$ (10,000 shuffles, $p < 0.05$) for more than 30% of the time points during the optogenetic stimulation. A spine qualified as responsive if optogenetically-evoked transients were detected in at least one out of the five sessions. Spines for which no significant optogenetically-evoked transients were detected in any of the five sessions were classified as unresponsive.

For the analysis presented in Fig. 4f, g, the spines were sorted into five categories according to their maximal fluorescence transient during optogenetic stimulation ranging from 'lowest' (in the lowest 25th percentile of all spines) to low (between the 25th and 50th percentile of all spines), mid (between the 50th and 75th percentile of all spines), high (between 75th and 95th percentile of all spines) and highest (above 95th percentile of all spines).

**Morphological analysis.** All spines were visually inspected and annotated in the projection images from the successive imaging session. Spines were then classified into four categories depending on the survival sequences, as follows:

- Formed if appearing in one of the sessions and staying until the end of experiments.
- Eliminated if present since the first session but disappearing during the experiments without reappearing in any of the following sessions.
- Persistent if present on all recording days.
- Transient if appearing and disappearing during the experiment.

The turnover ratio was calculated as the sum of formed and eliminated spines divided by the number of spines present between the days. The survival fraction at a given day was calculated as the fraction of spines present in the corresponding imaging session that were present in all preceding imaging sessions from the beginning.

Spine densities were obtained by dividing the number of spines present on dendrites by the length of the dendritic segment.

**Spine volume.** The volume of each spine was estimated as previously reported[7,31]. Briefly, each t-projection was deconvolved using the plugins Diffraction PSF 3D and Iterative Deconvolve 3D[62] (Fiji). All pixels in a spine ROI whose brightness exceeded the mean fluorescence of the background were summed and divided by the average fluorescence of the same number of pixels taken from the closest dendrite ROI. Since the resulting number is a ratio, it is unit-free and spine volume is given in arbitrary units.

For estimation of spine volume on the stacks obtained at the isosbestic excitation wavelength, a z-projection was computed from the registered frames (StackReg, Fiji) excluding frames with excessive motion followed by the same strategy as described above.

**Spatial spine distribution.** The interspine distances were calculated as previously reported[31]. Briefly, the distance between two spines was calculated as the dendritic length between the base of a pair of spines. The mass centers of the spines were projected on the dendrite. A path was then obtained using Dijkstra's algorithm and the resulting distance was converted to micrometers based on a previous field of view measurement on the microscope.

**Cell count.** Using CellPose[63], cells in the red channel of z-stacks of CA3b confocal images, around the injection site in which ChrimsonR was expressed, were segmented with the Segment Anything model.

Automatically generated ROIs were visually inspected and manually curated. To estimate the number of ChrimsonR-expressing CA3 cells, the density of cells per stack was multiplied by the estimated volume of CA3 (-0.4 mm³). An average of cell densities across all slices was then obtained for each mouse.

**Statistics.** Statistical analyses were performed using custom scripts on Python or R Studio (Foundation for Statistical Computing). A custom Python script was used to perform bootstrap analysis with 10,000 random permutations of the data to generate a null distribution. Data points were considered significant if they exceeded 95% of the values from the shuffled distribution ($p < 0.05$).

Pearson's chi-square tests (function *chisq.test*, stats library) were used to test the significance in case of categorical data (Fig. S3), an exact binomial test (function *binom.test*, stats library) was performed to test for the fraction of responsive spines (Fig. 2a) and linear mixed models (function *lmer*, lmerTest library) were performed using R Studio. Dendrites were qualified as random effects as indicated in the legend for each test. Pairwise comparisons were corrected with Bonferroni correction.

### Reporting summary

Further information on research design is available in the Nature Portfolio Reporting Summary linked to this article.

## Data availability

Source data are provided with this paper. The imaging data generated in this study have been deposited at the G-Node GIN Repository under accession code https://gin.g-node.org/SW_lab/Rais_Wiegert_Nat_Commun_2026. Source data are provided with this paper.

## Code availability

No standalone code was generated in this study. Custom modifications of acquisition and analysis scripts are available directly from the authors.

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

## Acknowledgements

We thank Maxime Maheu for help with data analyses, Stefan Schillemeit and Kathrin Sauter for excellent technical support, Wei Yang for initial training on spine imaging and Michael D. Adoff and Daniel A. Dombeck for advice on cranial window surgery. Ingke Braren of the UKE Vector Facility produced rAAV vectors. This work was supported by European Research Council (ERC2016-StG-714762 to J.S.W.) and the German Research Foundation, DFG (project numbers: 273915538 - SPP1926 and 278170285 - FOR2419/P6, to J.S.W.).

## Author contributions

C.R. and J.S.W. designed the experiments and wrote the manuscript. C.R. performed the experiments and analyzed the data. J.S.W. acquired funding and supervised the study.

## Funding

## Competing interests

The authors declare no competing interests.
