## [Transparent Peer Review file · Nature Communications]

Functional Synaptic Connectivity Shapes Spine Stability in the Hippocampus

Corresponding Author: Professor J. Simon Wiegert

Version 0:

Reviewer comments:

Reviewer #1

(Remarks to the Author)

This paper reported that the relation between spine stability and calcium response overtime in CA1 in hippocampus for awake mice while repeatedly stimulating presynaptic contralateral CA3 pyramidal neurons with optogenetics. They revealed that functionally active spines tended to form clusters of strongly connected spines along the same dendrite and are more stable than non-responsive spines. The text is written well and the results are interesting. This reviewer has some comments.

It is interesting to know whether the post-synaptic spine calcium response would nicely correlate with pre-synaptic glutaminergic releasing response using glutamate sensitive dye iGluSnFR3 expression. I know that it is very difficult experiment. If it would be impossible, then I would suggest that the authors should cite a paper published by Dr. Sasaki, J Physiology, 2012;590(19):4869-80, doi: 10.1113/jphysiol.2012.237685, at the sentence in page 4, line 131-132.

Page 5. Fig. 2e, f

e) Color code in this figure panel seems indicating distance between two spines, but it is not stated clearly neither in the figure legend nor main body. It is difficult to understand the figure in this condition. Explain the figure clearly to make readers understand it. Add scale bar.

f) It is hard to distinguish lines in the panel, because the difference of the lines is so faint, and color of the lines, dark blue, black, dark gray, seems quite similar, which should be clear. Perhaps sky blue, red, light orange and black would be better, for instance. I personally could not identify the lines for the unresponsive spines and/or shuffled.

Page 6. Fig. 3d, e

There is no clear explanation for "Turnover ratio of unresponsive spine" and "Turnover ratio of responsive spines", although the authors explained "Turnover ratio". State it clearly in the main body. Additionally, I would like to suggest to add "in a dendritic segment" at the end of figure legend in Fig. 3d)

Page 8. Line 332

Cite this paper after the last sentence in the discussion: Sohn et al, Science Advances 2022. It had nicely proved what the authors mentioned in the last sentence.

Reviewer #2

(Remarks to the Author)

In this manuscript Rais and Wiegert tackle to key issue of the relationship between the persistence and functional activation of dendritic spines. To this aim they use two-photon optical microscopy - to track activity as well as size of dendritic spines located on dorsal CA1 hippocampal pyramidal neurons - and direct optogenetic stimulation of contralateral dorsal hippocampal CA3 - to control the activation of a subset of dorsal CA1 hippocampal dendritic spines -. All of this in awake, head-fixed mice spontaneously moving on a treadmill. Their results show a positive correlation between spine size - estimated by GCaMP brightness - activation - measured by the relative increase in GCaMP brightness during optogenetic stimulation - and persistence over 4 days. In addition, they report clustering of active spines on basal dendrites. This work builds on previous ex- and in vivo studies addressing temporal dendritic spine stability and crucially adds a well-

controlled measurement of spine activation. In fact, the major strength of this work lays in the ability to control the activation of single spines deep in the brain of live mice and to track this activity over extended periods of time.

The findings are timely, interesting and well presented. Overall, this work will be significant for scientists interested in the relationship between structural plasticity and activity.

I do not have any major concern but a few minor comments.

- As the relationship between activity and size is key, it would be important to show some more details in Figure 2. For instance the actual distributions in addition to the cumulative ones in Fig. 2d and the Pearson or Spearman correlation values in Fig. 2d.

- Authors should specify the N of the statistics in the figure legends especially when in the same figure N can be spines or dendrites (as in Figs. 1 and 3).

- There are some significant conclusions based on data showed in Supp Fig. 7 and 8, the authors should consider move these data to main figures.

- Line 75, The sentence starting with "We further confirmed" seems to be somewhat cut.

- Line 84, "Shaffer" should be "Schaffer"

Reviewer #3

(Remarks to the Author)

Summary

The authors address an important question and describe an intuitive and interesting result: strong functional CA1-CA3 synapses (i.e. those that are activated by CA3 contralateral somatic stimulation) are more likely to be physically stable over time. While the study presents intriguing data on the relationship between synaptic activation and structural stability, substantial revisions are needed. Addressing the methodological concerns, clarifying experimental details and statistical methods, and providing a more detailed discussion of both evoked and natural activity will significantly enhance the manuscript's impact. With these improvements, the work could provide a more compelling contribution to our understanding of synaptic stability in the hippocampus.

Comments

They're only stimulating contralateral CA3, and only a small part of it (not all CA3 subregions i.e. CA3a, b, and c and the d-v axis). They do not describe this well in the paper and it is critical to know in order to correctly interpret the synaptic level results. Because they are not in control of all the potential functional synapses, the vast majority of synapses they are recording from they do not have functional control over. It is unclear if the authors are claiming that synapses not evoked by their opto-stim are generally weaker and therefore more likely to turn over because they are not functional? I think this is what the authors are indeed claiming. But, unresponsive spines are only non-functional in the sense that they are not connected to the neurons being opto-stimulated, they may still be functionally connected to non-stimulated CA3 neurons. Am I missing something here? If I am not, then the authors can only really compare the weaker versus stronger responsive synapses, as they have no clue what the other synapses are connected to and have no ability to drive them to say whether they are functional and what strength they have.

The authors do not look at the natural activation of spines. All the work is focused on the stimulated spines, but of course spines are activated naturally during behavior. Why not look at their natural activation dynamics and relate that to physical and functional stability?

Their opto-stim'd during immobility to limit motion artefacts but the CA1 behaves very differently during motion versus immobility (during motion: theta oscillations, interneurons are highly active, place cells are activated, limited or no replays, limited SWRs, dopamine VTA inputs ramp up if rewards are present, LC inputs are activated during motion onset etc). It would be very revealing to see how the results differ during motion when the CA1 is in a very different state as this state could modulate how spines are activated through modulation of dendritic excitability.

The authors measure the dendritic distance between spines that respond to their opto-stim and find them to be a bit closer to one another than they are to non-responsive spines. I find this hard to wrap my head around. Does it mean that CA3 somas that are closer together are more likely to form synapses on CA1 dendrites that are closer together, because they are only opto-stimulating a subpopulation of CA3 cells that are close to one another?

Also, being a bit closer does not mean they are clustered. Clustering is when spines are adjacent to one another on the same dendrite and can therefore electrically and molecularly influence each other to activate non-linear dendritic events (spikes) or plasticity (through cooperativity). Their data as it stands in Fig. 2 do not show clustering, but the conclusions by the author suggest this is how they interpret the data.

The authors state in the Discussion session that non-responsive spines in the basal dendrites could be from other inputs onto these dendrites. They mention EC and amygdala, but EC inputs do not form synapses on CA1 basal dendrites (they're restricted to the distal tuft dendrites in SLM). Also, Amygdala inputs are only found in ventral hippocampus, and not dorsal where the spine imaging was done.

I think the authors should also discuss how many contralateral CA3 somas they think they are stimulating in their protocol, and where they are located. It is not all of contralateral CA3, which seems to be the assumption. The expression of the opsin, the depth of light penetration, the accessibility to ventral regions, the extent in which they are hitting CA3a, b, or c, the transfection of the virus, the efficiency in which somas can be triggered to fire with this opsin...all these factors will determine

which sub-population of contralateral CA3 somas they are able to activate. This should be discussed. Also, they should have images of opsin expression which could help inform the subpopulation they are stimulating. This should be included in the paper.

Due to the imaging method, spines may be activated but not be detected, as calcium imaging is an indirect measure of synapse activation. The authors should state their ideas on what it is they are measuring with calcium imaging. What is the source of calcium influx, can spines be activated and not produce calcium influx? If so, how should we think of these missing events? Could a responsive spine in one session turn into a non-responding spine because even though the synapse is activated it fails to cause calcium influx?

The imaging session and treadmill session lengths are not noted.

The imaging location is briefly mentioned in Fig. S1c, but the authors do not describe in detail the inclusion criteria for the imaging plane location, including distance from soma and dendritic branch order, both of which contribute to dendritic spine morphogenesis and connectivity.

Titer amount for viruses is missing

None of the images contain scale bars

Sample sizes are largely missing:

- Fig. 2 notes the sample sizes including the number of spines, dendrites, and mice. However, it is unclear if this is the same sample set as in Fig. 3, which includes no sample size information.
- Fig. S3 does not include sample sizes.
- In Fig. S5, for S5a, the sample size states "N = 8 dendrites" but does not describe which group has 8 dendrites and what the sample size is for the other group. Also, in S5b-e, no sample sizes are noted.
- Fig. S6 does not include sample sizes.
- Fig. S7 does not include sample sizes.

The statistics of this manuscript are largely incomplete or missing, making many of the conclusions difficult to fully substantiate:

- In Fig 1o and 2c-f, they do not define the statistical tests.
- In Fig S3b, it is unclear which comparison is significant. The authors state, "most spines responded only in one session", but the statistics in the figure are unclear as to which comparisons are significant.
- In Fig. S5c, the statistical results seem to be missing for responsive versus unresponsive.
- In Fig. S5e, it is unclear which groups are not significant and which groups are significant. The figure legend also does not include the p-value.
- Multiple figure panels do not define the statistical tests used in the respective figure legend.
- The authors very briefly mention "dendrites and mice were qualified as random effects when appropriate." However, they do not state in the results or figure legends when this statement applied to their statistical tests. Especially with large, repeated measurements (i.e. hundreds of dendritic spine measurements) from the same sample (i.e. dendrite and/or brain), linear mixed models can be more appropriate. Please see the publication by Wilson and colleagues (DOI: 10.1016/j.jneumeth.2017.01.003) for a more detailed explanation.

Error bar definitions are missing for figure panels S5b-e.

The following additional data analysis could strengthen the conclusions and provided additional, interesting observations:

- It is well known that running and exercise influences many neuronal properties, including dendritic spines. It would be interesting to correlate the treadmill running data during the sessions with the dendritic spine responsive and unresponsive grouped data.
- In the conclusion (lines 287-291), the authors describe the responsive spines typically only respond in one session, suggesting synaptic reconfiguration across time. In their last figure panel 3g, the authors show spine volume is significant positive correlation between volume and number of responding sessions. Additional analysis to compare the other data sets across days would greatly strengthen the conclusions and innovation of this manuscript.
- In figure S7, it would be helpful to compare the volume across all 4 categories in one graph, such as using a bar graph of each category with individual points with volume on the y-axis.

Additional explanation in the results section describing how they calculated the responsive spines vs unresponsive spines (i.e. 30% of the time points are greater than 95% of shuffled) would add clarification when the authors first describe the "responding spines".

It is unclear why the authors completed the imaging sessions during a passive treadmill task if they only optogenetically stimulated during periods of immobility. Further explanation of this at the beginning of the results would help.

Clarification of the imaging session length will be relevant to the interpretation of the persistent, transient, formed, and eliminated dynamic spine data. Spine can form and be eliminated on the scale of minutes. Especially since the authors paired the imaging with a passive treadmill task, there may be significant differences within imaging sessions as well as across imaging sessions. Changes across imaging sessions results may be skewed if they were compared at the different time points during the session compared to just the start or beginning of each treadmill and imaging session. Additional clarification at which time point during the session the images were taken would provide clarity. Also, measuring the spine dynamics within each imaging session would additionally strengthen the manuscript.

- In the methods, the mice age ranges from 2 to 9 months. Dendritic spine density is typically drastically different between 2 and 9 months. This large age range could lead to muddled results and increased variability compared to if the authors

picked a narrower window. Further analysis demonstrating the vast age difference does not alter the conclusions would be beneficial. Additionally, completing additional animals in a smaller age window would strengthen the conclusions.

- Fig 3d-e and S6 may be more clearly understood as a bar graph comparing responsive versus unresponsive and grouped by session day, with individual data points as points similar to fig 3b.
- In line 124, the authors state that unresponsive spines “never” respond in the session. However, their responsive spine criteria include responding at greater than 30% of the time points. Thus, a spine could respond at 29% of the time points and not be considered a responsive spine. Clearer wording in line 124 would help clarification.
- The authors state a “slightly higher proportion” (line 130) regarding fig S3c. However, fig S3c does not include any statistical tests. Statistical tests are needed to determine a significant difference. Additionally, if there is a difference, further explanation on this interesting phenomena of increased activity at the first imaging session would strengthen the manuscript.
- In line 233-234 the results state, “the survival fraction of all initially identified spines at each imaging session.” However, the methods state, “The survival fraction is calculated as the fraction of spines present on the given that were already existing on the first day of the experiment.” Corrected wording in the methods and consistency about the reference point (each imaging session vs first day of the experiment) are both necessary.
- Inconsistencies in the results and figures that should be addressed include:
 - o In Fig 3b the figure says “frequency” but the results text says “proportion”.
 - o The figure legend states that Fig 3f colors match Fig 3b. However, the colors and the order of the colors do not match. Correct colors and clear labels are needed.
 - o Spine volume should be in a designated unit instead of arbitrary units.

In the first sentence of the discussion, the authors state “the stability of glutamatergic spines at hippocampal CA3-CA1 synapses is dependent on connectivity strength” regarding their optogenetically determined responsive spines. However, a large fraction of the unresponsive spines also remain stable across the imaging sessions, suggesting that although the responsive CA3-CA1 synapses may have a higher level of persistence, and thus stability, it is not dependent on this connectivity. Clearer wording would more accurately describe this conclusion.

Also relevant for the Discussion, CA1 basal dendritic and spine imaging during behavior and the relevance of dendritic signals to place field formation and stability across time was done in the Dombeck Lab and is highly relevant to this work, but not mentioned or referenced:

Sheffield, M.E., Adoff M. D., Dombeck D.A. Increased prevalence of calcium transients across the dendritic arbor during place field formation. *Neuron* 96, 490-504 (2017).

Sheffield, M.E. & Dombeck, D.A. Calcium transient prevalence across the dendritic arbour predicts place field properties. *Nature* 517, 200-204 (2015).

Version 1:

Reviewer comments:

Reviewer #1

(Remarks to the Author)

Dear Authors,

I like the revision. It is much clearer than the original manuscript and message is clear.

Very minor comment:

Figure 1 legend

A) - G) should be a) - g). I mean they should be small letter.

Reviewer #2

(Remarks to the Author)

I still stand by my previous statement that "The findings of this work are timely, interesting and well presented. Overall, this work will be significant for scientists interested in the relationship between structural plasticity and activity."

The authors have fully addressed my concerns in the revised version of their manuscript, thus I fully support publication.

Reviewer #3

(Remarks to the Author)

The authors have addressed all my comments.

REVIEWER COMMENTS

Reviewer #1:

This paper reported that the relation between spine stability and calcium response overtime in CA1 in hippocampus for awake mice while repeatedly stimulating presynaptic contralateral CA3 pyramidal neurons with optogenetics. They revealed that functionally active spines tended to form clusters of strongly connected spines along the same dendrite and are more stable than non-responsive spines. The text is written well and the results are interesting. This reviewer has some comments.

It is interesting to know whether the post-synaptic spine calcium response would nicely correlate with pre-synaptic glutaminergic releasing response using glutamate sensitive dye iGluSnFR3 expression. I know that it is very difficult experiment. If it would be impossible, then I would suggest that the authors should cite a paper published by Dr. Sasaki, J Physiology, 2012;590(19):4869-80, doi: 10.1113/jphysiol.2012.237685, at the sentence in page 4, line 131-132.

Thank you for the positive feedback on our manuscript. Indeed, the question regarding the reliability of calcium imaging to detect synaptic transmission is important. Our lab has used this method previously to investigate synaptic function in an all-optical way. Therefore, we wanted to convince ourselves that spine calcium transients indeed report synaptic transmission in a reliable way. For this, we had performed a pilot experiment in hippocampal slices earlier (see Reviewer Fig. 1). We expressed SF-iGluSnFR(A184S) together with jRGECO1a in postsynaptic CA1 neurons for simultaneous imaging of glutamate release and spine calcium transients. Presynaptic CA3 neurons expressed ChR2 and hM4D (for chemogenetic inhibition of synaptic transmission). We stimulated presynaptic CA3 neurons optogenetically under high release probability conditions and saw a high correspondence between glutamate and calcium transients. Inhibition of synaptic release by applying CNO reduced both glutamate and calcium responses. This experiment indicates that calcium transients reliably report synaptic transmission in spines of CA1 pyramidal neurons.

In addition to these experiments, an in vivo study from the Losonczy lab showed a similar correspondence between glutamate release and postsynaptic calcium events in CA1 spines. See figure 3a in DOI: 10.1038/s41586-024-08325-9.

We have added new text to the Results and Discussion sections to better explain the relationship between EPSCaTs and synaptic transmission (page 4, line 131 ff):

“EPSCaTs depend on depolarization by AMPA receptors²² and they are largely mediated by NMDA receptors²³ with contributions from voltage-gated calcium channels²⁴ and intracellular stores in some cases²⁵. EPSCaTs are indicative of synaptic strength^{26,27} and they were recently shown to reliably report glutamatergic synaptic transmission at CA1 pyramidal neurons in vivo²⁸. Thus, EPSCaTs serve as a good proxy for postsynaptic responses at spines of CA1 pyramidal neurons. “

As suggested, we have also mentioned the contribution of presynaptic calcium fluctuations in the Discussion section (where we now also cite the paper by Sasaki et al., page 13, line 432 ff):

“Notably, responsive spines exhibited EPSCaTs primarily in a single imaging session, suggesting dynamic functional reconfiguration of connectivity across sessions. This reconfiguration may reflect changes in synaptic connectivity strength, wherein both spontaneous fluctuations in presynaptic AP-evoked transmitter release⁵⁰ and fluctuations in postsynaptic calcium responses could contribute to the transient loss of detectable EPSCaTs at weaker synapses.”

Reviewer Figure 1. Simultaneous, dual-color imaging of glutamate release and calcium transients in CA1 pyramidal neurons of organotypic hippocampal slices. A) Schematic outline of the experiment. Chr2 is expressed together with the chemogenetic tool hM4Di in presynaptic CA3 neurons (a presynaptic bouton is shown in gray). SF-iGluSnFR(A184S) is co-expressed with jRGECO1a in postsynaptic CA1 neurons via single-cell electroporation (the postsynaptic spine is shown in teal). Blue-light stimulation of CA3 cells via an optical fiber elicits synaptic transmission. Application of CNO activates hM4Di, which suppresses synaptic release. **B)** Left: two-photon images of a dendritic segment of a CA1 cell expressing SF-iGluSnFR(A184S) and jRGECO1a (green and red, respectively). Right column shows fluorescence changes of both indicators in a single trial. Note the spatially confined signal, indicating local glu release at a single spine which corresponds to a postsynaptic, localized calcium transient. Right: Fluorescence traces ($\Delta F/F_0$) of multiple trials in high release probability conditions (4 mM extracellular Ca^{2+}) in absence (left) or presence (right) of CNO. **C)** Quantification of SF-iGluSnFR(A184S) and jRGECO1a peak responses shown in B). **D)** Correlation analysis of SF-iGluSnFR(A184S) and jRGECO1a. Note the high correlation between glutamate signals and spine Ca^{2+} transients.

Page 5. Fig. 2e, f

e) Color code in this figure panel seems indicating distance between two spines, but it is not stated clearly neither in the figure legend nor main body. It is difficult to understand the figure in this condition. Explain the figure clearly to make readers understand it. Add scale bar.

A sentence to clarify the color code has been added in the legend of the figure: *“The color code indicates the distance between the example spine and all other spines present along the dendritic segment.”*

Scale bars have been added to all images.

f) It is hard to distinguish lines in the panel, because the difference of the lines is so faint, and color of the lines, dark blue, black, dark gray, seems quite similar, which should be clear. Perhaps sky blue, red, light orange and black would be better, for instance. I personally could not identify the lines for the unresponsive spines and/or shuffled.

Apologies for the non-optimal choice of colors. We have changed the colors, the line type and the arrangement of lines in this and other panels for better readability.

Page 6. Fig. 3d, e

There is no clear explanation for “Turnover ratio of unresponsive spine” and “Turnover ratio of responsive spines”, although the authors explained “Turnover ratio”. State it clearly in the main body. Additionally, I would like to suggest to add “in a dendritic segment “ at the end of figure legend in Fig. 3d)

The sentence has been modified for better clarity: “First, we separately assessed the turnover ratio for responsive and unresponsive spines in every dendritic segment. For both groups (responsive or unresponsive spines), the turnover ratio is defined as the sum of spine formation and spine elimination rates between consecutive imaging sessions.”

Page 8. Line 332

Cite this paper after the last sentence in the discussion: Sohn et al, Science Advances 2022. It had nicely proved what the authors mentioned in the last sentence.

The reference has been added.

Reviewer #2:

In this manuscript Rais and Wiegert tackle to key issue of the relationship between the persistence and functional activation of dendritic spines. To this aim they use two-photon optical microscopy - to track activity as well as size of dendritic spines located on dorsal CA1 hippocampal pyramidal neurons - and direct optogenetic stimulation of contralateral dorsal hippocampal CA3 – to control the activation of a subset of dorsal CA1 hippocampal dendritic spines -. All of this in awake, head-fixed mice spontaneously moving on a treadmill. Their results show a positive correlation between spine size – estimated by GCaMP brightness – activation – measured by the relative increase in GCaMP brightness during optogenetic stimulation – and persistence over 4 days. In addition, they report clustering of active spines on basal dendrites.

This work builds on previous ex- and in vivo studies addressing temporal dendritic spine stability and crucially adds a well-controlled measurement of spine activation. In fact, the major strength of this work lays in the ability to control the activation of single spines deep in the brain of live mice and to track this activity over extended periods of time.

The findings are timely, interesting and well presented. Overall, this work will be significant for scientists interested in the relationship between structural plasticity and activity.

I do not have any major concern but a few minor comments.

Thank you for the positive evaluation of our manuscript. Below, please find our detailed point-by-point responses.

- As the relationship between activity and size is key, it would be important to show some more details in Figure 2. For instance the actual distributions in addition to the cumulative ones in Fig. 2d and the Pearson or Spearman correlation values in Fig. 2d.

As suggested, for additional details, we added a new supplementary figure (Figure S4) that shows the actual distribution of spines.

Fig. S4. The volume of responsive spines is higher than the volume of unresponsive spines. a) Distribution of the volume of responsive (blue) and unresponsive (grey) spines as in Figure 2c. 22 bins equally spaced (20 a.u.) were used. **b)** Volumes of all responsive and unresponsive spines. Permutation tests. *: $p < 0.05$.

As suggested, we now indicate the Pearson correlation value for Fig 2d in the legend: "Pearson correlation: 0.27."

- Authors should specify the N of the statistics in the figure legends especially when in the same figure N can be spines or dendrites (as in Figs. 1 and 3).

Apologies for not providing this information more consistently. We have added more detailed information on statistics in all figure legends.

- There are some significant conclusions based on data showed in Supp Fig. 7 and 8, the authors should consider move these data to main figures.

We followed this suggestion and present the main information from these two supplemental figures in a more intuitive and condensed way in the new main figure 4. We still left some additional, more detailed information in the supplement (Figure S8 and S9).

- Line 75, The sentence starting with “We further confirmed” seems to be somewhat cut.

We rephrased the sentence for better clarity (page 3, line 95 ff): *“By injecting five adult mice with ChrimsonR in cICA3 and non-conditional jRCaMP8m in iICA1 for dense labelling, we further confirmed that optogenetically evoked synaptic input to iICA1 was maintained constant across imaging sessions (Figure S2).”*

- Line 84, “Shaffer” should be “Schaffer”

We corrected this typo.

Reviewer #3:

Summary

The authors address an important question and describe an intuitive and interesting result: strong functional CA1-CA3 synapses (i.e. those that are activated by CA3 contralateral somatic stimulation) are more likely to be physically stable over time. While the study presents intriguing data on the relationship between synaptic activation and structural stability, substantial revisions are needed.

Addressing the methodological concerns, clarifying experimental details and statistical methods, and providing a more detailed discussion of both evoked and natural activity will significantly enhance the manuscript's impact. With these improvements, the work could provide a more compelling contribution to our understanding of synaptic stability in the hippocampus.

Thank you for your fair and critical assessment of our work, which was very helpful. We sincerely apologize for the lack of clarity regarding the statistics and methods, as well as the missing information in the text and figures. We have significantly revised the manuscript to improve its quality. Please see our responses to the individual points below. We also recognize that we failed to adequately explain certain concepts and presented vague interpretations of some results. To address these concerns, we have performed additional analyses, included new figure panels, and rewritten sections of the manuscript. Your critical feedback was very helpful, and we hope that our manuscript has improved sufficiently to be published in Nature Communications.

Comments

They're only stimulating contralateral CA3, and only a small part of it (not all CA3 subregions i.e. CA3a, b, and c and the d-v axis). They do not describe this well in the paper and it is critical to know in order to correctly interpret the synaptic level results. Because they are not in control of all the potential functional synapses, the vast majority of synapses they are recording from they do not have functional control over. It is unclear if the authors are claiming that synapses not evoked by their opto-stim are generally weaker and therefore more likely to turn over because they are not functional? I think this is what the authors are indeed claiming. But, unresponsive spines are only non-functional in the sense that they are not connected to the neurons being opto-stimulated, they may still be functionally connected to non-stimulated CA3 neurons. Am I missing something here? If I am not, then the authors can only really compare the weaker versus stronger responsive synapses, as they have no clue what the other synapses are connected to and have no ability to drive them to say whether they are functional and what strength they have.

We apologize for not explaining this better. As you correctly pointed out, we can only detect spines that receive input from optogenetically activated presynaptic CA3 cells. Spines that do not respond may represent weak synapses that do not cross the detection threshold (see below for more details), or they may belong to synapses with non-stimulated presynaptic neurons. Non-responding spines belong to a mixed pool of stimulated subthreshold synapses and non-stimulated synapses, which show a wide distribution of strengths. Therefore, we likely even underestimate the differences between weak and strong synapses (assuming that the pool of non-stimulated synapses also contains a fraction of strong synapses).

We rewrote parts of the manuscript and included new paragraphs to better explain this (page 6, lines 189 ff).

“However, because unresponsive spines may also include strong synapses that were not activated, we next restricted our analysis to responsive spines to avoid confounds associated with synapses of unknown activation status. We tested whether spine volume correlated with the amplitude of excitatory postsynaptic calcium transients (EPSCaTs). Indeed, we observed a positive correlation between spine volume and EPSCaT amplitude (Fig. 2d), consistent with previous findings that larger spines correspond to stronger synapses³³. Thus, although unresponsive spines likely represent a heterogeneous population with a broad distribution of synaptic strengths, our results indicate that responsive spines preferentially correspond to strong synapses.

To ensure that our spine volume estimates were not influenced by intracellular calcium levels, we acquired volumetric stacks at an isosbestic (calcium independent) excitation wavelength (Figure S5a). We observed a strong correlation between the two measures (Figure S5b), indicating that both strategies yielded comparable volume estimates. In addition, we also controlled for the relationship between the volume obtained from the time-series for all spines and calcium fluctuations (Figure S5c). We did not find a significant correlation, suggesting that volume measurements were not confounded by dynamic brightness changes arising from intracellular calcium dynamics. Finally, we asked whether spine volume was directly related to the presence of calcium responses within individual imaging sessions. To address this, we subdivided all responsive spines into sessions in which a calcium response was detected and sessions in which no response was observed, and compared the corresponding spine volumes. We found no significant effect of response presence on measured spine volume, although we observed a weak trend toward slightly larger volumes during responding sessions (Fig. S5d). Thus, while spine volume correlated with EPSCaT amplitude across spines, this relationship was only weakly expressed within individual spines across sessions, likely reflecting a non-linear relationship between calcium transients, synaptic strength, and spine morphology.”

We still argue that spines showing calcium transients represent the most reliable and most stable synapses, as further illustrated by the new figure panel 4e, which illustrates a positive

correlation between EPSCaT amplitude and the number of session where responses were detected. Furthermore, to test the relationship between synaptic weight and spine stability more directly, we grouped all spines into 5 categories according to their maximal GCaMP amplitudes (including all sub-threshold spines). In this way we avoid categorization into responsive vs. unresponsive spines, but rather we categorize them according to the strength of their fluorescence transients, as suggested. Indeed, this analysis revealed a strong relationship between the maximal $\Delta F/F$ amplitude and the spine lifetime. This new analysis is now shown in figure 4f,g.

See page 11, line 353 ff: *“To establish this relationship between synaptic strength, spine volume and spine stability more firmly, we asked if the frequency of synaptic responses (i.e. the number of imaging sessions in which spines showed EPSCaTs) is related to EPSCaT amplitude and spine head size. We found a significant positive correlation between the frequency of responses and the average volume of responding spines as well as EPSCaT amplitude (Figure 4d, e).*

Because our categorization into responsive and unresponsive spines could have led to the exclusion of weakly responsive spines, we considered again all imaged spines and split them into five categories according to their highest maximum fluorescent transients across all sessions, including all spines that were previously categorized as unresponsive (Figure 4f). This alternative analysis confirmed the relationship between the maximal calcium response and the spine lifetime (Figure 4g). Thus, our data demonstrate a close relationship between spine size, functional connectivity and spine stability at hippocampal CA3-CA1 synapses in vivo.”

The authors do not look at the natural activation of spines. All the work is focused on the stimulated spines, but of course spines are activated naturally during behavior. Why not look at their natural activation dynamics and relate that to physical and functional stability? They opto-stim'd during immobility to limit motion artefacts but the CA1 behaves very differently during motion versus immobility (during motion: theta oscillations, interneurons are highly active, place cells are activated, limited or no replays, limited SWRs, dopamine VTA inputs ramp up if rewards are present, LC inputs are activated during motion onset etc). It would be very revealing to see how the results differ during motion when the CA1 is in a very different state as this state could modulate how spines are activated through modulation of dendritic excitability.

Indeed, an ideal experiment would associate structural spine turnover with natural synaptic activity. However, there are several challenges that make these experiments extremely difficult, if not impossible, in our hands.

First, natural activity is low during periods of quiescence and motion artifacts are too strong to reliably monitor single spines during locomotion in our chronic preparation. This is now documented in Fig S1 f-g. The hippocampus is surrounded by the third and lateral ventricles and therefore, it is subject to intrinsic motion when the animal is moving. Thus, even if the skull is motion-free, there is some residual intrinsic motion of deep brain structures. In fact, we are aware of only very few studies that managed recordings of spine activity during locomotion^{1,2}. However, these studies did not monitor spines across imaging session, but rather were limited to a single session. Our attempts to achieve a chronic preparation where we stabilized the hippocampus so that we could monitor single spines both functionally and structurally over many days without motion artefacts during running failed. For example, stabilizing the hippocampus with refractive index-matched polymer gel under the window deteriorated optical resolution so that spines could not be optimally resolved. Moreover, using a deeper window, which would “push down” on the hippocampus to prevent its motion during running would result in significant alterations of the morphology, such as flattening, which we aimed to avoid.

Figure S1g-h. Motion-related analysis. **g)** Example trace of dendritic fluorescence including bouts of locomotion. Calcium events are marked in pink. Average z-projections are shown for baseline fluorescence (right, green) and for a dendritic calcium transient (magenta, top right). Motion events led to out-of-focus signal (left inset, gray frame). **h)** Quantification of frame shift during immobility and motion. Top: Permutation test. Arrow represents the true difference.

Second, natural activity is often associated with dendritic calcium transients due to firing of the postsynaptic neuron. Therefore, even if we perfectly controlled for motion artefacts, it is difficult to relate calcium transients at single spines to synaptic input.

Future studies should indeed aim to overcome the limitations of motion and record synaptic activity and spine structure over many days or weeks. However, the fact that such a study does not yet exist for the hippocampus indicates that this is not trivial and requires novel approaches that are beyond the scope of this study.

To acknowledge the importance of monitoring naturally active spines, we now added a paragraph to the discussion highlighting the future need for chronic recordings of functionally identified synapses active during natural behavior (page 13, lines 445 ff):

“Another strategy to circumvent artefacts associated with plasticity-inducing stimulation protocols is to rely on sensory stimuli⁵³ or to pair naturally occurring synaptic input with postsynaptic manipulation of neuronal activity^{47,54}. Recent studies have employed postsynaptic optogenetic stimulation of CA1 cells to induce behavioral time scale plasticity followed by functional measurements of naturally occurring synaptic transmission^{47,55}. While these studies linked synaptic plasticity to cellular function in vivo, they did not follow the functional or structural properties of these synapses over time. Thus, future functional and structural studies of identified spines associated with a specific behavior or a cellular function over extended time periods will be important to shed light on the role of individual synapses in long-term information processing and storage in the brain. Indirect evidence suggests that spine stability is increased at CA1 neurons, which are part of an engram⁵⁶. Directly assessing the lifetime of spines involved in the control of these neurons and concurrent animal behavior would provide critical insights into the synaptic mechanisms underpinning memory encoding and storage within the neuronal networks involved in episodic memory formation and storage⁵⁷.”

The authors measure the dendritic distance between spines that respond to their opto-stim and find them to be a bit closer to one another than they are to non-responsive spines. I find this hard to wrap my head around. Does it mean that CA3 somas that are closer together are more likely to form synapses on CA1 dendrites that are closer together, because they are only opto-stimulating a subpopulation of CA3 cells that are close to one another?

We do not think that this is due to neighboring CA3 cells converging on close-by dendritic areas of CA1 cells. Rather, axons from individual CA3 cells typically form a small number of synapses on a given postsynaptic CA1 cell. These synapses are not randomly distributed across the dendritic branches of CA1 cells, but tend to be proximal to each other in the dendritic area where the axonal collateral is in contact with the postsynaptic CA1 cell³⁻⁶. We mention this now in the manuscript and also have rephrased the corresponding section (page 6, line 214 ff):

“Moreover, due to their highly divergent connectivity, CA3 neurons form only a small number of synapses with a given CA1 neuron. These synapses are often found in close proximity to each other, given that they arise from the same presynaptic axon branch³⁵. To assess whether responsive spines were in closer proximity to each other compared to all other spines, we measured all-to-all dendritic distance between the spines in the field of view (Figure 2e). Our comparison of the spatial distribution of responsive and all other spines revealed a closer spatial proximity (shorter dendritic length) between responsive spines versus unresponsive spines (Figure 2f). This suggests – as expected from CA3-CA1 connectivity – that functional inputs from a defined population of pre-synaptic neurons are more likely to be in spatially closer proximity rather than randomly distributed³⁶.

Regarding our estimate of the population of stimulated CA3 neurons, please see our response further below.

Also, being a bit closer does not mean they are clustered. Clustering is when spines are adjacent to one another on the same dendrite and can therefore electrically and molecularly influence each other to activate non-linear dendritic events (spikes) or plasticity (through cooperativity). Their data as it stands in Fig. 2 do not show clustering, but the conclusions by the author suggest this is how they interpret the data.

We agree. The term “clustering” was misleading and is incorrect in our case (see response above). We replaced it for the term “proximity”. We also rewrote the Discussion section to be more specific about the putative role/function of proximal inputs and the limitations that we face in detecting true clusters. We hope that this now clarified.

Page 12, line 388 ff: *“The larger head volume of responsive spines compared to unresponsive neighboring spines indicates a stronger synaptic connection^{7,31,32} with pre-synaptic cICA3 neurons. Those spines could belong to functional ensembles, which previously underwent LTP^{40,41} as suggested by a previous report⁴². We observed a slightly higher spatial proximity between strong, responsive synapses, which may indicate that small groups of spines receive input from the same presynaptic CA3 cells or from CA3 cells with similar functional tuning. These strong inputs may have contributed to postsynaptic calcium responses in CA1 neurons during our initial screening (Fig. 1c-e) by contributing sufficient depolarization to reach cellular spike threshold⁴¹. Notably, the proximity of responsive spines in our study does not represent anatomical clusters. This is likely due to the fact that we stimulated only a subset of all CA3 cells providing direct synaptic input to the postsynaptic CA1 cell. Thus, only a fraction of all spines on a given dendritic segment receives direct, optogenetically evoked input, while other synapses on the same dendrite may originate from non-stimulated CA3 cells, which remain elusive during postsynaptic calcium imaging. Functional clusters might only be revealed by activity of all presynaptic neurons that share functional properties, as is the case during spatial navigation where spatially tuned CA3 cells are co-active in a given dendritic location^{43,44}. Nonetheless, the closer spatial proximity of responsive spines aligns with findings from those recent studies demonstrating that spine clusters on place cells were correlated with the spatial tuning of these neurons during a spatial navigation task⁴⁵. As a consequence, such clustering of functionally related spines likely contributes to neuronal firing within specific place fields, consistent with prior studies that have linked functional spine clustering to learning^{31,34,43,44}. Due to their increased stability, such synaptic populations can form long-lasting ensembles with a feature-specific*

presynaptic population of CA3 cells. This helps to stabilize specific synaptic input to a feature-selective CA1 cell (e.g., place cell firing) over time despite fluctuations at individual synapses.”

The authors state in the Discussion session that non-responsive spines in the basal dendrites could be from other inputs onto these dendrites. They mention EC and amygdala, but EC inputs do not form synapses on CA1 basal dendrites (they're restricted to the distal tuft dendrites in SLM). Also, Amygdala inputs are only found in ventral hippocampus, and not dorsal where the spine imaging was done.

That's right. We apologize for this confusing statement, which was left over from an earlier version of the manuscript. The mention of amygdala and entorhinal cortex has been removed.

I think the authors should also discuss how many contralateral CA3 somas they think they are stimulating in their protocol, and where they are located. It is not all of contralateral CA3, which seems to be the assumption. The expression of the opsin, the depth of light penetration, the accessibility to ventral regions, the extent in which they are hitting CA3a, b, or c, the transfection of the virus, the efficiency in which somas can be triggered to fire with this opsin...all these factors will determine which sub-population of contralateral CA3 somas they are able to activate. This should be discussed. Also, they should have images of opsin expression which could help inform the subpopulation they are stimulating. This should be included in the paper.

As suggested, we included a new section in the manuscript, providing more information on this aspect (page 3, line 70 ff): “We estimated the light distribution in area CA3 using a Monte Carlo simulation⁷ considering light scattering and absorption at 633 nm and the optical properties of the light fiber (200 μ m, 0.22 NA). Given the modeled distribution of light intensities and the spike thresholds for ChrimsonR⁸, we assume that we reliably stimulated the entire cICA3, including distal and medial regions. Most of the distal and medial CA3 cells project to stratum oriens⁹. Therefore, we expect a large portion of their axons to terminate in the stratum oriens of CA1 – also on the ipsilateral side, where we imaged. In fact, Schaffer commissural axons from contralateral CA3 make up approx. 50% of the synaptic input and they preferably terminate in stratum oriens of the ipsilateral CA1¹⁰. Therefore, by imaging spines in stratum oriens of iICA1, we expect to reliably detect postsynaptic responses to optogenetic stimulation of cICA3.”

We added the Monte Carlo simulation to better estimate the light propagation and ChrimsonR activation in CA3 to the updated fig. S1. In addition, we provide an estimate of the number of CA3 cells excited in each mouse based on the number of ChrimsonR expressing CA3 cells in brain slices. We used CellPose to segment these cells and, subsequently, to extrapolate the total number of ChrimsonR-expressing CA3 cells that were illuminated with suprathreshold irradiance levels.

We also updated fig. S1 with an overview image of the contralateral CA3. We often also detected expression in DG. However, this is not a major concern, since DG does not project to the contralateral side. A similar expression pattern was obtained in a previous study from the Deisseroth lab, which used a similar approach to ours to stimulate synaptic input in ipsilateral CA1 via optogenetic activation of contralateral CA3 (see fig S6 in DOI: [10.1016/j.cell.2022.12.035](https://doi.org/10.1016/j.cell.2022.12.035)).

Due to the imaging method, spines may be activated but not be detected, as calcium imaging is an indirect measure of synapse activation. The authors should state their ideas on what it is they are measuring with calcium imaging. What is the source of calcium influx, can spines be activated and not produce calcium influx? If so, how should we think of these missing events? Could a responsive spine in one session turn into a non-responding spine because even though the synapse is activated it fails to cause calcium influx?

This is an important point and we did not explain it well enough in the manuscript. Indeed, excitatory postsynaptic calcium transients (EPSCaTs) were used in multiple previous studies to estimate synaptic transmission at CA1 pyramidal neurons. We have added new sections to the manuscript, explaining the choice of calcium imaging and the validity of the approach (page 4, line 131 ff): “EPSCaTs depend on depolarization by AMPA receptors²² and they are largely mediated by NMDA receptors²³ with contributions from voltage-gated calcium channels²⁴ and intracellular stores in some cases²⁵. EPSCaTs are indicative of synaptic strength^{26,27} and they were recently shown to reliably report glutamatergic synaptic transmission at CA1 pyramidal neurons in vivo²⁸. Thus, EPSCaTs serve as a good proxy for postsynaptic responses at spines of CA1 pyramidal neurons.”

Furthermore, we have done a pilot experiment in hippocampal slices where we simultaneously monitored calcium transients and glutamate release at single spines of CA1 neurons (please also see response to reviewer 1 and **Reviewer Figure 1**). We found that calcium transients reliably reported glutamate release at those synapses and that no glutamate release was detected at synapses that did not display calcium transients.

Thus, EPSCaTs serve as a useful and reliable proxy for synaptic transmission in the hippocampus.

The imaging session and treadmill session lengths are not noted.

We added the following statements to the Method section:

Page 15, line 510: “Mice were placed on a 2m-long linear treadmill...”.

Page 15, line 544 ff: “The lengths of imaging sessions depended on the number of dendrites to image as well as the periods of spontaneous running activity of the mouse. The session durations were as follows: “

	Mouse #1	Mouse #2	Mouse #3	Mouse #4	Mouse #5
Day 0	133 min	70 min	36 min	54 min	45 min
Day 4	95 min	71 min	22 min	44 min	38 min
Day 8	75 min	64 min	19 min	44 min	34 min
Day 12	54 min	61 min	19 min	40 min	38 min
Day 16	60min	128 min	20 min	34 min	59 min

The imaging location is briefly mentioned in Fig. S1c, but the authors do not describe in detail the inclusion criteria for the imaging plane location, including distance from soma and dendritic branch order, both of which contribute to dendritic spine morphogenesis and connectivity.

Thanks for pointing this out. We added the following statement to the Methods section:

Page 15, line 553 ff: “For this, segments horizontal to the imaging plane on first or second order basal dendrites in close proximity of the soma were chosen. The global signal-to-noise ratio (SNR) from each dendritic segment was calculated as follows: (1) the signal amplitude was estimated by taking the 95th percentile minus the 5th percentile of the dendritic trace, (2) the noise was estimated by calculating the standard deviation of the high-pass filtered dendritic trace (cutoff: 10 Hz), (3) the signal amplitude was divided by the noise estimation. Dendritic segments with a signal-to-noise ratio below 8 for any of the recorded sessions were discarded (average global dendritic SNR: 16.2 and range: 8-44, average global spine SNR: 7.5 and range: 3-32).”

Titer amount for viruses is missing

More detailed information on the viruses (incl. titers) is now included in the Methods section as follows:

Page 14, line 479 ff: “0.3 μ L of AAV9-CaMKII-ChrimsonR-mScarlet-KV2.1 (1×10^{13} vg/mL, AddGene # 124651-AAV9) viral suspension was first injected in left CA3 (-2.0mm AP, -2.3mm ML, -2.5mm DV relative to Bregma) using a custom-made air-pressure driven injection system. For the spine recording experiments, 0.5 μ L of a 1:1 mix of AAV9-CaMKII-Cre (3×10^9 vg/mL, gift from D. Kuhl) and AAV1-Syn-flex-jGCaMP7b-WPRE ($1,4 \times 10^{13}$ vg/mL, AddGene #104493-AAV1) or AAV9-hsyn-DIO-GCaMP7b-P2A-GSG-mRuby3 ($8,15 \times 10^{13}$ vg/mL, AddGene #130900) was injected in right CA1 (-2.0mm AP, +1.5mm ML, -1.5mm DV relative to Bregma). In experiments where dense CA1 cellular recording were done, mice were instead injected with 0.5 μ L of AAV9-Syn-jGCaMP8m (4×10^{12} vg/mL, AddGene #162375-AAV9) in CA1.”

None of the images contain scale bars

Apologies for omitting the scale bars. They have been added to the images now.

Sample sizes are largely missing:

- Fig. 2 notes the sample sizes including the number of spines, dendrites, and mice. However, it is unclear if this is the same sample set as in Fig. 3, which includes no sample size information.

We see that our reporting of sample sizes was not very clear. Sorry for this. We now mention details of the sample size in the figure legends for each panel. We hope that this is clearer now.

- Fig. S3 does not include sample sizes.

Sample sizes have been added.

- In Fig. S5, for S5a, the sample size states “N = 8 dendrites” but does not describe which group has 8 dendrites and what the sample size is for the other group. Also, in S5b-e, no sample sizes are noted.

We clarified this with the following statement: “Proportion of different dynamic spine categories on dendrites in control animals (no optogenetic stimulation, N= 8 dendrites) and optogenetically-stimulated animals (N=17 dendrites).”

- Fig. S6 does not include sample sizes.

Sample sizes have been added.

- Fig. S7 does not include sample sizes.

Sample sizes have been added.

The statistics of this manuscript are largely incomplete or missing, making many of the conclusions difficult to fully substantiate:

- In Fig 1o and 2c-f, they do not define the statistical tests.

Fig 1o reports simply the total number of spines that were imaged across all animals per session. No statistical test was done here. A statistical comparison of the fraction of responding spines per dendrite is shown in 1p. Here, we added additional information on the statistics.

The information on statistical tests for figure 2c-f is now added to the legend.

- In Fig S3b, it is unclear which comparison is significant. The authors state, “most spines responded only in one session”, but the statistics in the figure are unclear as to which comparisons are significant.

We added matrices to the figure panels showing the results of the pairwise Chi-square tests.

- In Fig. S5c, the statistical results seem to be missing for responsive versus unresponsive.

The statistical test has been added (now fig. S7c).

- In Fig. S5e, it is unclear which groups are not significant and which groups are significant. The figure legend also does not include the p-value.

The figure has been modified to better indicate which groups were compared. The legend now includes the p-values (now fig. S7e).

- Multiple figure panels do not define the statistical tests used in the respective figure legend.

The statistical tests are now indicated in all figure legends.

- The authors very briefly mention “dendrites and mice were qualified as random effects when appropriate.” However, they do not state in the results or figure legends when this statement applied to their statistical tests. Especially with large, repeated measurements (i.e. hundreds of dendritic spine measurements) from the same sample (i.e. dendrite and/or brain), linear mixed models can be more appropriate. Please see the publication by Wilson and colleagues (DOI: 10.1016/j.jneumeth.2017.01.003) for a more detailed explanation.

Thank you for this suggestion. We have now replaced all statistical tests by a linear mixed model, except for Chi-square tests and permutation tests.

Error bar definitions are missing for figure panels S5b-e.

Error bar definitions (“Mean \pm s.e.m.”) are now added.

The following additional data analysis could strengthen the conclusions and provided additional, interesting observations:

- It is well known that running and exercise influences many neuronal properties, including dendritic spines. It would be interesting to correlate the treadmill running data during the sessions with the dendritic spine responsive and unresponsive grouped data.

Treadmill running occurred only sporadically and we did not record it continuously throughout the sessions. As explained below and in the Methods, we used the treadmill to detect voluntary motion and to avoid motion artefacts. While it is certainly very interesting to assess the influence of exercise on spine dynamics, more systematic experiments are required to properly address this question, which are beyond the aims of this study. As also explained below, spine turnover within a single session is extremely low. Thus, effects of exercise on spine turnover will likely only become apparent on longer time intervals. However, this means that exercise also needs to be tracked systematically for each mouse in the home cage, which would be an interesting question for a follow-up study.

- In the conclusion (lines 287-291), the authors describe the responsive spines typically only respond in one session, suggesting synaptic reconfiguration across time. In their last figure panel 3g, the authors show spine volume is significant positive correlation between volume and number of responding sessions. Additional analysis to compare the other data sets across days would greatly strengthen the conclusions and innovation of this manuscript.

We followed the suggestion and added several new analyses to the figures. In figure 4e we now also observe a strong significant correlation between the overall EPSCaT amplitude and the number of sessions during which a spine responded. This indicates that spines with large responses participate more reliably in synaptic communication over time. Notably, this is still true if we assume that failure to detect responses in spines with smaller EPSCaTs may be due to our limited detection threshold. Being below detection threshold means that synaptic transmission is either absent or weak. In turn, spines with larger EPSCaT amplitudes display a longer lifetime as shown now in fig. 4g.

Finally, in figure S5d we compared the volumes of all responsive spines in sessions where they showed responses to session where they did not show responses. This analysis confirms that detection of EPSCaTs did not affect our spine volume measurements per se.

- In figure S7, it would be helpful to compare the volume across all 4 categories in one graph, such as using a bar graph of each category with individual points with volume on the y-axis.

We added a new panel, as suggested, to figure 4 (previously fig S7, now fig. 4b).

Additional explanation in the results section describing how they calculated the responsive spines vs unresponsive spines (i.e. 30% of the time points are greater than 95% of shuffled) would add clarification when the authors first describe the “responding spines”.

We rewrote this section and also added the following explanation to the Methods section (page 16, line 592 ff): *“Permutations were made to find statistically significant time points during the optogenetic stimulation window. To minimize false-positive spines due to noise in the signal, as for dendrites, a spine was classified as responding if the calcium signal was greater than 95% of the values of the mean- shuffled $\Delta F/F$ (10,000 shuffles, $p < 0.05$) for more than 30% of the time points during the optogenetic stimulation.”*

It is unclear why the authors completed the imaging sessions during a passive treadmill task if they only optogenetically stimulated during periods of immobility. Further explanation of this at the beginning of the results would help.

The following explanation for the use of a passive treadmill has been added to the Methods section (page 15, lines 511 ff): *“Using a treadmill, we could better monitor motion that can lead to out-of-focus signal, discarding those frames for analysis while maximizing the number of trials without motion artefacts. Moreover, a treadmill was chosen, since this provides the animals with the opportunity to voluntarily move during the imaging session and therefore is less stressful compared to full immobilization.”*

We also updated the corresponding Results section (page 3, line 81 ff): *“During two-photon spine imaging, we recorded the motion of head-fixed mice using a passive treadmill, enabling closed-loop optogenetic stimulation selectively during periods of immobility, when motion artefacts were largely absent (Figure 1b). Accurate, continuous monitoring of locomotion via the treadmill was essential to prevent brain-motion artefacts that can lead to out-of-focus signals (Figure S1g-h). This closed-loop approach allowed us to exclude motion-affected frames while maximizing the number of trials without motion artefacts.”*

Clarification of the imaging session length will be relevant to the interpretation of the persistent, transient, formed, and eliminated dynamic spine data. Spine can form and be eliminated on the scale of minutes. Especially since the authors paired the imaging with a passive treadmill task, there may be significant differences within imaging sessions as well as across imaging sessions. Changes across imaging sessions results may be skewed if they were compared at the different time points during the session compared to just the start or beginning of each treadmill and imaging session. Additional clarification at which time point during the session the images were taken would provide clarity. Also, measuring the spine dynamics within each imaging session would additionally strengthen the manuscript.

This is indeed an important point, which we addressed in detail in our previous work. In Yang et al. (2021, DOI: 10.1371/journal.pbio.3001146.), we analyzed dendritic spine turnover in head-fixed, awake mice, including during wakefulness on a treadmill (see Fig. 8 and Supplementary Fig. 10). To our knowledge, this study provided the first characterization of CA1 spine dynamics during wakefulness. We found that CA1 spine turnover over the course of one hour was very low and remained stable throughout the imaging sessions. Importantly, fast spine dynamics within imaging sessions did not differ between awake and anesthetized conditions, indicating that the imaging procedure itself did not measurably affect spine turnover.

Additionally, in our experiments shown in the present manuscript, the sequence of dendritic imaging was randomized in each session in order to mitigate putative cumulative effects of optogenetic stimulation during the session. The Methods section has been modified to point this out more clearly (page 15, lines 542): *“At the beginning of each session, for each mouse, the order of imaged dendrites was shuffled to account for potential influence of the optogenetic stimulation across the session”*.

- In the methods, the mice age ranges from 2 to 9 months. Dendritic spine density is typically drastically different between 2 and 9 months. This large age range could lead to muddled results and increased variability compared to if the authors picked a narrower window. Further analysis demonstrating the vast age difference does not alter the conclusions would be beneficial. Additionally, completing additional animals in a smaller age window would strengthen the conclusions.

We previously addressed this question systematically for CA1 pyramidal neurons (Yang et al., 2021). In that study, we analyzed dendritic spine turnover in the hippocampus over extended time periods (several months), allowing us to directly assess potential age-related effects on spine density and spine dynamics between 3 and 9 months of age (see Fig. S10A,C in Yang et al.). We found no differences in spine survival rate, turnover ratio, or spine density between younger and older animals during repeated awake imaging sessions. These results indicate that age is unlikely to be a major confounding factor for spine dynamics in the present study.

Please also note that the range of 2-9 months refers to the entire procedure (i.e., the age at the surgery until the end of the last experiment). The youngest mouse was 10 weeks old at the time of surgery and thus 14 weeks (> 3 months) old at the first imaging session.

- Fig 3d-e and S6 may be more clearly understood as a bar graph comparing responsive versus unresponsive and grouped by session day, with individual data points as points similar to fig 3b.

Following your suggestion, we modified the plots in the figures showing the comparison of responsive and non-responsive spines for each session as scatter plots with individual data points and lines connecting subgroups of the same dendrites.

Figure 3d-e. Turnover ratio and survival rate. **d)** Turnover ratio of responsive spines versus unresponsive spines in a dendritic segment between two sessions. Pairs of filled and open circles connected by a line represent responsive and unresponsive spines on a given dendrite. Linear Mixed Model with dendrites as random effect and pairwise comparisons with Bonferroni correction. Mean \pm s.e.m. **e)** Survival fraction of responsive spines versus unresponsive spines at each session. Pairs of filled and open circles connected by a line represent responsive and unresponsive spines on a given dendrite. Linear Mixed Model with dendrites as random effect and pairwise comparisons with Bonferroni correction. Large circles: mean \pm s.e.m.

- In line 124, the authors state that unresponsive spines “never” respond in the session. However, their responsive spine criteria include responding at greater than 30% of the time points. Thus, a spine could respond at 29% of the time points and not be considered a responsive spine. Clearer wording in line 124 would help clarification.

Indeed our 30% criterion was used to minimize false-positive timepoints due to noise. According to this criterion, responses in at least 3 out of 10 trials needed to be above the significance level during optogenetic stimulation. Thus, the expression “never” is indeed not correct. We rephrased the corresponding sections in the manuscript and explain the classification in more detail in the Methods (page 17, line 593): “To minimize false-positive spines due to noise in the signal, as for dendrites, a spine was classified as responding if the calcium signal was greater than 95% of the values of the mean- shuffled $\Delta F/F$ (10,000 shuffles, $p < 0.05$) for more than 30% of the time points during the optogenetic stimulation. A spine qualified as responsive if optogenetically-evoked transients were detected in at least one out of the five sessions. Spines for which no significant optogenetically-evoked transients were detected in any of the five sessions were classified as unresponsive.”

To further account for the concern that spines classified as unresponsive might still receive sub-threshold input, which escaped our detection, and therefore display increased lifetimes, we analyzed spine lifetime as a function of their maximal GCaMP transient amplitude. As already indicated in our response to the first point (see above) we found a direct positive relationship between the maximum response recorded and the lifetime of these spines, suggesting that the strongest spines have the longest lifetimes. Thus, even though some weakly responding spines might be included in our “unresponsive” cohort, this will not change the overall interpretation that (strongly) responsive spines displayed higher stability.

- The authors state a “slightly higher proportion” (line 130) regarding fig S3c. However, fig S3c does not include any statistical tests. Statistical tests are needed to determine a significant difference. Additionally, if there is a difference, further explanation on this interesting phenomena of increased activity at the first imaging session would strengthen the manuscript.

Apologies for not presenting this in sufficient detail. We added a comprehensive matrix with pairwise chi-square tests to the figure panel. The first session (day 0) shows a higher fraction of spines responding for the first time. This is expected and explained by the experimental strategy. We explain this now also in the main text, as suggested (page 5, lines 158 ff):

“A majority of spines showed a first response in the first imaging session (approx. 1/3 of all spines ever showing a response). Accordingly, spines showing EPSCaTs for the first time were also found in later sessions, with an equal distribution across those sessions (Figure S3c). The bias towards the first session is inherent to the experimental approach: by default, any responding spine in this session is classified as responding for the first time, since no preceding sessions exist. In addition, since we screened for dendrites with active spines in the first session, this may also have contributed to a higher fraction. In the following sessions, the same dendrites were revisited irrespective of spine responses. Nevertheless, the variability in responses (i.e., spines not responding in all sessions or even functionally emerging in later session) suggests that functional connectivity gets reconfigured between sessions and therefore, individual synaptic inputs drift over time.”

- In line 233-234 the results state, “the survival fraction of all initially identified spines at each imaging session.” However, the methods state, “The survival fraction is calculated as the fraction of spines present on the given that were already existing on the first day of the experiment.” Corrected wording in the methods and consistency about the reference point (each imaging session vs first day of the experiment) are both necessary.

The sentence now reads as follows: *“To explore this in more detail, we assessed the survival fraction defined as those spines detected in a given imaging session that were present in all preceding imaging sessions from the beginning.”*

We also modified the Methods section, accordingly.

- Inconsistencies in the results and figures that should be addressed include:
 - o In Fig 3b the figure says “frequency” but the results text says “proportion”.

We modified the figure for the consistency.

- o The figure legend states that Fig 3f colors match Fig 3b. However, the colors and the order of the colors do not match. Correct colors and clear labels are needed.

Clearer labels have been added and the legend has been modified accordingly.

- o Spine volume should be in a designated unit instead of arbitrary units.

The spine volume is estimated by dividing the mean fluorescence of the spine head with the mean fluorescence of the dendrites, thus there is no unit for spine volume. We added a corresponding sentence to the Methods section.

In the first sentence of the discussion, the authors state “the stability of glutamatergic spines at hippocampal CA3-CA1 synapses is dependent on connectivity strength” regarding their optogenetically determined responsive spines. However, a large fraction of the unresponsive spines also remain stable across the imaging sessions, suggesting that although the responsive CA3-CA1 synapses may have a higher level of persistence, and thus stability, it is not dependent on this connectivity. Clearer wording would more accurately describe this conclusion.

Good point. We changed this statement to: *“In this study, we demonstrate that the stability of glutamatergic spines at hippocampal CA3-CA1 synapses is regulated by connectivity strength.”*

Moreover, we completely rewrote the discussion for better clarity and to address several additional points raised above.

Also relevant for the Discussion, CA1 basal dendritic and spine imaging during behavior and the relevance of dendritic signals to place field formation and stability across time was done in the Dombeck Lab and is highly relevant to this work, but not mentioned or referenced:
Sheffield, M.E., Adoff M. D., Dombeck D.A. Increased prevalence of calcium transients across the dendritic arbor during place field formation. *Neuron* 96, 490-504 (2017).
Sheffield, M.E. & Dombeck, D.A. Calcium transient prevalence across the dendritic arbour predicts place field properties. *Nature* 517, 200-204 (2015).

That's right. These references have been added.

References

1. Adoff, M. D. *et al.* The functional organization of excitatory synaptic input to place cells. *Nat. Commun.* **12**, 1–15 (2021).
2. Gonzalez, K. C. *et al.* Synaptic basis of feature selectivity in hippocampal neurons. *Nature* 1–9 (2024) doi:10.1038/s41586-024-08325-9.
3. Bartol, T. M., Jr *et al.* Nanoconnectomic upper bound on the variability of synaptic plasticity. *eLife* **4**, e10778 (2015).
4. Parajuli, L. K. & Koike, M. Three-Dimensional Structure of Dendritic Spines Revealed by Volume Electron Microscopy Techniques. *Front. Neuroanat.* **15**, (2021).
5. Bloss, E. B. *et al.* Single excitatory axons form clustered synapses onto CA1 pyramidal cell dendrites. *Nat. Neurosci.* **21**, 353–363 (2018).
6. Sorra, K. E. & Harris, K. M. Occurrence and three-dimensional structure of multiple synapses between individual radiatum axons and their target pyramidal cells in hippocampal area CA1. *J. Neurosci.* **13**, 3736–3748 (1993).
7. Stujenske, J. M., Spellman, T. & Gordon, J. A. Modeling the Spatiotemporal Dynamics of Light and Heat Propagation for In Vivo Optogenetics. *Cell Rep.* **12**, 525–534 (2015).
8. Klapoetke, N. C. *et al.* Independent optical excitation of distinct neural populations. *Nat. Methods* **11**, 338–346 (2014).
9. Martig, A. K. & Mizumori, S. J. Y. Place Cells. in *Encyclopedia of Behavioral Neuroscience* (eds Koob, G. F., Moal, M. L. & Thompson, R. F.) 70–78 (Academic Press, Oxford, 2010). doi:10.1016/B978-0-08-045396-5.00154-8.
10. Martin, S. J., Shires, K. L. & da Silva, B. M. Hippocampal Lateralization and Synaptic Plasticity in the Intact Rat: No Left–Right Asymmetry in Electrically Induced CA3-CA1 Long-Term Potentiation. *Neuroscience* **397**, 147–158 (2019).

REVIEWERS' COMMENTS

Reviewer #1 (Remarks to the Author):

Dear Authors,

I like the revision. It is much clearer than the original manuscript and message is clear.

Very minor comment:

Figure 1 legend

A) - G) should be a) - g). I mean they should be small letter.

Thank you. We fixed the panel labels in the figure legend of Figure 1 in the revised manuscript.

Reviewer #2 (Remarks to the Author):

I still stand by my previous statement that "The findings of this work are timely, interesting and well presented. Overall, this work will be significant for scientists interested in the relationship between structural plasticity and activity."

The authors have fully addressed my concerns in the revised version of their manuscript, thus I fully support publication.

Thank you.

Reviewer #3 (Remarks to the Author):

The authors have addressed all my comments.

Thank you for the critical feedback.